# Patch-based Memory Gate Model in Time Series Foundation Model

## Abstract

Recently reconstruction-based deep models have been widely used for time series anomaly detection, but as their capacity and generalization capability increase, these models tend to over-generalize, often reconstructing unseen anomalies accurately. Prior works have attempted to mitigate this by incorporating a memory architecture that stores prototypes of normal patterns. Nevertheless, these approaches suffer from high training costs and have yet to be effectively integrated with time series foundation models (TSFMs). To address these challenges, we propose **MOMEMTO**, an improved TSFM variant for anomaly detection, enhanced with a patch-based memory module to mitigate over-generalization. The memory module is designed to capture representative normal patterns from multiple domains and enables a single model to be jointly fine-tuned across these domains through a multi-domain training strategy. MOMEMTO initializes memory items with latent representations from a pre-trained encoder, organizes them into patch-level units, and updates them via an attention mechanism. We evaluate our method using 23 univariate benchmark datasets. Experimental results demonstrate that MOMEMTO, as a single model, achieves higher scores on AUC and VUS metrics compared to baseline methods, and further enhances the performance of its backbone TSFM, particularly in few-shot learning scenarios.

## 1 Introduction

Complex cyber-physical systems operate continuously, accumulating vast volumes of sequential time series data from multiple sensors. Monitoring these systems and detecting anomalies at an early stage is highly beneficial in terms of operational cost. In real-world anomaly detection scenarios, it is common to simultaneously handle time series from multiple domains that originate from different application areas (e.g., distinct sensor types or process configurations). Such scenarios frequently involve data imbalance and limited labeled data. We formulate this problem as an unsupervised learning task to tackle these challenges.

Reconstruction-based deep models are widely used for unsupervised time series anomaly detection. In this approach, an encoder-decoder network is trained with a self-supervised objective to reconstruct input sequences. These models aim to reproduce normal patterns precisely, while generating higher reconstruction errors for anomalies. Various models have been proposed within this paradigm, including OmniAnomaly (Su et al., 2019), Anomaly Transformer (Xu et al., 2022), TranAD (Tuli et al., 2022), and TimesNet (Wu et al., 2023). Recent models have achieved strong performance due to the representation learning capability of neural networks. However, existing reconstruction-based approaches often suffer from over-generalization, where anomalous inputs are reconstructed with high accuracy (Gong et al., 2019; Park et al., 2020). This issue arises when the encoder captures distinctive features of anomalies or when the decoder has sufficient capacity to reconstruct abnormal representations.

Recent research has increasingly focused on TSFMs, which are pre-trained on large-scale time series data from a variety of sources. Their strong generalization capability enables TSFMs to perform robustly in few-shot and zero-shot settings on forecasting tasks. However, applying TSFMs to anomaly detection presents two main challenges. First, most TSFMs, such as MOIRAI (Woo et al., 2024), TimesFM (Das et al., 2024), Chronos (Ansari et al., 2024), Time-MoE (Shi et al., 2024), and Sundial (Liu et al., 2025) are primarily

designed for time series forecasting. They often employ decoder-only architectures that limit their suitability for detection tasks. Second, unlike in time series forecasting, the generalization capability of TSFMs leaves them vulnerable to over-generalization under reconstruction approaches, paradoxically hindering their effectiveness as a unified model across diverse domains.

To address these challenges, we redesign the gate memory module of MEMTO (Song et al., 2023), a reconstruction-based deep model for time series anomaly detection. This module stores memory items that represent the prototypical features of normal patterns. When anomalies are reconstructed using these normal memory items, the resulting outputs resemble normal samples, thus alleviating over-generalization. However, MEMTO is highly sensitive to memory initialization, requiring encoder pre-training to obtain informative and stable representations. Given MEMTO's one-model-per-domain design, this sensitivity leads to high training cost when handling multiple domains. It also uses a point-level memory rather than a patch-based structure, which reduces its effectiveness in detecting interval or periodic anomalies (Shen, 2025).

Furthermore, we adopt the pre-trained encoder from MOMENT (Goswami et al., 2024), a TSFM trained on large-scale time series corpora. MOMENT employs a patch-level masked representation learning strategy, where input sequences are divided into fixed-length patches and trained to reconstruct the masked patches. Its encoder, built on the T5 architecture (Raffel et al., 2020), can be coupled with task-specific decoders, providing highly transferable time series representations and flexibility across diverse tasks. However, like other TSFMs, MOMENT adopts multi-domain pre-training, but as a reconstruction-based model for anomaly detection, it still remains vulnerable to over-generalization. This limitation prevents it from fully exploiting the potential benefits of multi-domain fine-tuning across diverse domains.

Building upon previous methods, we introduce **MOMEMTO**, an improved variant of MOMENT adapted for reconstruction-based time series anomaly detection. The model integrates the pre-trained encoder from MOMENT with the patch-based memory module. Unlike MEMTO, our model benefits from MOMENT's encoder, which already possesses sufficient representational capacity. This provides well-initialized features for the memory module and alleviates MEMTO's sensitivity to memory initialization. The patch-based memory module stores encoder outputs that are organized at the patch level. Whereas the original memory items stored prototypical features of normal patterns restricted to a single domain, the patch-based memory items are designed to capture representative normal patterns from multiple domains. This design alleviates over-generalization while enabling the model to learn effectively in diverse domains.

Specifically, multi-domain training refers to jointly fine-tuning a single model across domains from diverse application areas, rather than training a separate model for each domain. This strategy facilitates knowledge sharing across domains and allows more effective use of limited data, while also reducing computational cost in both training time and memory usage. Under the multi-domain training strategy, a single MOMEMTO model achieves strong results on 23 univariate benchmark datasets compared to baseline methods. In few-shot learning scenarios, it demonstrates clear improvements over its backbone TSFM. Our contributions can be summarized as follows:

- We introduce a patch-based memory architecture to support TSFMs, which stores features from multiple domains and is compatible with various patch-based methods.

- MOMEMTO extends MOMENT into a TSFM specialized for univariate time series anomaly detection, supporting multi-domain training and leveraging the patch-based memory module to mitigate over-generalization.

- MOMEMTO achieves higher AUC and VUS scores than the evaluated baselines on 23 univariate benchmark datasets and improves over its MOMENT backbone in few-shot settings.

## 2 Related Work

**Transformers and patching for time series**  Transformers (Vaswani et al., 2017) have demonstrated strong performance on sequential data processing. For time series analysis, Transformers benefit from the self-attention mechanism and are used to capture reliable long-range temporal dependencies (Kitaev et al.,

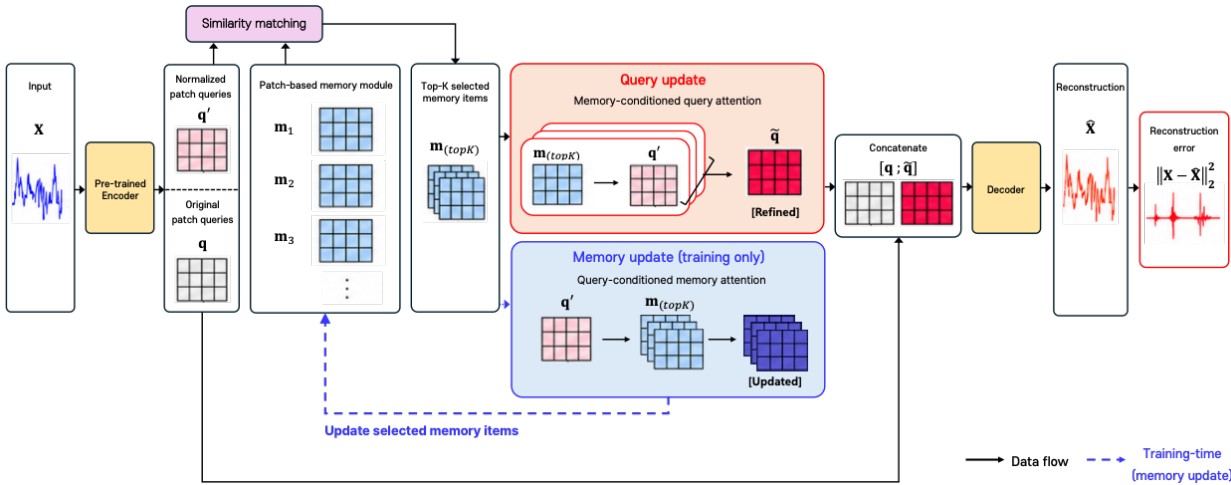

Figure 1: Architecture of MOMEMTO.

2020; Li et al., 2021; Zhou et al., 2021; Liu et al., 2024). Among them, PatchTST (Nie et al., 2023) is a Transformer-based model specifically designed for time series analysis. Its first strategy is patching, in which input sequences are segmented into subsequence-level patches, preserving local semantic information while reducing computational complexity. Its second strategy is a channel-independent processing scheme, where each univariate series is processed independently with shared Transformer weights, minimizing inter-channel interference and effectively capturing distinct temporal dynamics. Together, these designs enhance both the efficiency and the representation capability of Transformers for time series tasks.

**Memory-guided deep models**   Recently, memory architectures have been introduced to enhance neural models by enabling external storage and retrieval of long-term information. They have been applied in diverse domains such as natural language processing, including large-scale language modeling (Lample et al., 2019), retrieval-augmented generation (Lewis et al., 2020), and long-context understanding (Wang et al., 2023), and in computer vision tasks like video captioning (Lei et al., 2020), video object segmentation (Oh et al., 2019) as well as in reinforcement learning for episodic memory and sample-efficient decision making (Le et al., 2021). For anomaly detection, several approaches in computer vision employ memory modules to store features of normal patterns (Zhou et al., 2023). MemAE (Gong et al., 2019) integrates a memory module into an autoencoder. MNAD (Park et al., 2020) introduces a weighted memory-update strategy to capture temporal anomalies in videos. Based on these concepts, MEMTO is the first model to introduce a memory architecture into time series anomaly detection, employing a data-driven update mechanism. More recently, H-PAD (Shen, 2025) extends this idea by learning hybrid scale- and period-level prototypes to capture both interval and periodic anomalies.

MEMTO and H-PAD are representative memory-based approaches for reconstruction-based time series anomaly detection, but their memory mechanisms are not designed for the context of modern TSFMs. Both methods train their point-based backbones from scratch, update all memory components under a one-model-per-domain paradigm, and rely on additional embedding-level anomaly scoring beyond reconstruction error. Consequently, their memory quality largely depends on the representation quality of the backbone, making initialization and updates less stable. Unlike these existing methods, the proposed patch-based memory module leverages a pre-trained patch-based backbone and domain-adaptive selective memory updates to improve memory stability while keeping anomaly scoring simple through reconstruction error. This design is intended to support scalable multi-domain anomaly detection while maintaining compatibility with the patch-based TSFMs. To the best of our knowledge, this is the first attempt to adapt memory-guided reconstruction to TSFM-based time series anomaly detection.

## 3 Methodology

### 3.1 MOMEMTO

Figure 1 illustrates the overall architecture of MOMEMTO, which mainly consists of an encoder–decoder with the patch-based memory module. The input time series $\mathbf{x}$ is first processed by the encoder. The encoder output representation $\mathbf{q} \in \mathbb{R}^{P \times d_{model}}$ is L2-normalized to obtain $\mathbf{q}'$, which is used as queries to interact with the memory module, where $d_{model}$ denotes the embedding dimension. The normalized query vectors are compared with the stored memory items to retrieve the most relevant ones, and the selected memory items are subsequently updated conditioned on the queries. The queries are then refined via memory-based attention. The updated queries $\tilde{\mathbf{q}} \in \mathbb{R}^{P \times d_{model}}$ are combined with the original queries $\mathbf{q}$ (i.e., $[\mathbf{q}; \tilde{\mathbf{q}}] \in \mathbb{R}^{P \times 2d_{model}}$) and passed to the decoder. The decoder maps the combined query representation to the input space to reconstruct the time series $\hat{\mathbf{x}} \in \mathbb{R}^{P \times L}$.

#### 3.1.1 Pre-trained Encoder and Decoder

We adopt the pre-trained encoder from MOMENT-large as the backbone for our model. The encoder receives a univariate time series $\mathbf{x} \in \mathbb{R}^{N \times L}$ together with a mask vector of length $N$, where each entry is 0 or 1 to indicate unobserved and observed patches, respectively. Each input patch is treated as a token and embedded into a vector of length $d_{model}$. Patch-level self-attention is then applied to capture temporal dependencies and generate contextualized representations. For reconstruction, we use a shared lightweight decoder implemented as two fully-connected layers. This design mitigates the risk of over-generalization and promotes reliance on the encoder's representational capacity, while enabling the model to generalize across time series with varying sequence lengths.

#### 3.1.2 Patch-based Memory Module

To alleviate over-generalization in reconstruction-based models, we introduce the patch-based memory module as a redesigned variant of MEMTO's original memory module. Unlike MEMTO's point-level memory, it naturally aligns with the patch-based encoder representations of MOMENT. The module consists of $M$ memory items $\{\mathbf{m}_i\}_{i=1}^M$, where each memory item $\mathbf{m}_i \in \mathbb{R}^{N \times d_{model}}$ stores a prototypical normal pattern at the patch level. The module operates in a data-driven manner by retrieving relevant memory items, updating the selected memories, and refining the query representations through memory-based attention. Throughout this process, the memory items and queries are progressively aligned with representative normal patterns.

**Memory initialization** Each memory item $\mathbf{m}_i$ is initialized with a specific domain, which may encompass multiple datasets. It is stored as the mean of encoder representations from $S_i$ instances randomly sampled across the datasets belonging to that domain. These encoder representations are denoted by $\{\mathbf{q}_s\}_{s=1}^{S_i}$. The initial value of $\mathbf{m}_i$ is then computed as

$$\mathbf{m}_i = \frac{1}{S_i} \sum_{s=1}^{S_i} \mathbf{q}_s, \quad i = 1, \dots, M.$$

To ensure stable updates, memory items are consistently L2-normalized at the patch level.

**Memory alignment** For each input sequence, we identify the observed patches in the encoder output $\mathbf{q}$ using the input mask and retrieve the corresponding subset of representations. To ensure consistent interaction, each memory item $\mathbf{m}_i$ is aligned with the same subset of observed patches, and the queries are likewise L2-normalized.

$$\begin{array}{ccc} \mathbf{q} \in \mathbb{R}^{N \times d_{model}} & \xrightarrow{\text{retrieve}} & \mathbf{q} \in \mathbb{R}^{P \times d_{model}} \\ \mathbf{m}_i \in \mathbb{R}^{N \times d_{model}} & & \mathbf{m}_i \in \mathbb{R}^{P \times d_{model}} \end{array}$$

This alignment process allows the model to handle time series of varying lengths flexibly and selectively update memory items, enhancing adaptability across domains.

Table 1: Notation summary for MOMEMTO.

| Symbol | Shape/Type | Description |
|---|---|---|
| $L$ | scalar | Patch length |
| $P$ | scalar | Number of observed patches |
| $N$ | scalar | Maximum number of patches after zero-padding, with $P \leq N$ |
| $d_{\text{model}}$ | scalar | Encoder representation dimension |
| $M$ | scalar | Number of memory items |
| $K$ | scalar | Number of selected memory items used for update and retrieval |
| $\mathbf{x}$ | $P \times L$ | Input time series divided into observed patches |
| $\mathbf{q}$ | $P \times d_{\text{model}}$ | Original encoder output representation retained for the decoder input |
| $\mathbf{q}'$ | $P \times d_{\text{model}}$ | L2-normalized query used for memory interaction |
| $\mathbf{m}_i$ | $P \times d_{\text{model}}$ | $i$-th memory item after memory alignment |
| $\mathbf{m}_{(k)}$ | $P \times d_{\text{model}}$ | $k$-th selected memory item after memory alignment |
| $\mathbf{U}_\psi, \mathbf{W}_\psi$ | $d_{\text{model}} \times d_{\text{model}}$ | Learnable projection matrices in the memory update gate |
| $\mathbf{v}_{(k)}$ | $P \times d_{\text{model}}$ | Query-based representation used to update $\mathbf{m}_{(k)}$ |
| $\boldsymbol{\psi}_{(k)}$ | $P \times d_{\text{model}}$ | Element-wise gate for memory update |
| $\tilde{\mathbf{m}}_{(k)}$ | $P \times d_{\text{model}}$ | Updated candidate of the selected memory item |
| $w_{(k)}$ | scalar | Retrieval weight for the $k$-th selected memory item |
| $\tilde{\mathbf{q}}_{(k)}$ | $P \times d_{\text{model}}$ | Intermediate query representation from the $k$-th selected memory item |
| $\tilde{\mathbf{q}}$ | $P \times d_{\text{model}}$ | Final memory-refined query representation |
| $[\mathbf{q}; \tilde{\mathbf{q}}]$ | $P \times 2d_{\text{model}}$ | Concatenated representation passed to the decoder |
| $\hat{\mathbf{x}}$ | $P \times L$ | Reconstructed time series patches |

**Memory selection**   After memory alignment, the normalized query representation $\mathbf{q}'$ and the corresponding memory slices share the same patch-level structure. To select the most relevant memory items, we temporarily vectorize the aligned representations into $\text{vec}(\mathbf{q}'), \text{vec}(\mathbf{m}_i) \in \mathbb{R}^{Pd_{\text{model}}}$ and compute sequence-level similarities. The top-$K$ scores are then retrieved, and softmax normalization is applied only over these selected scores.

$$s_i = \frac{\text{vec}(\mathbf{q}')^\top \text{vec}(\mathbf{m}_i)}{\tau}, \qquad i = 1, \ldots, M,$$

$$\{s_{(k)}\}_{k=1}^K = \text{TopK}\left(\{s_i\}_{i=1}^M\right),$$

$$w_{(k)} = \frac{\exp(s_{(k)})}{\sum_{\ell=1}^K \exp(s_{(\ell)})}, \qquad k = 1, \ldots, K.$$

where $\tau$ is a temperature parameter, and $w_{(k)}$ is the retrieval weight assigned to the $k$-th selected memory item.

**Memory update**   From the memory selection step, we denote the memory items corresponding to the $K$ largest sequence-level similarities as $\{\mathbf{m}_{(k)}\}_{k=1}^K$. Rather than updating all memory items or relying on the initial domain assignment, the module updates only the retrieved memory items according to the current query. This allows the memory to adapt in a data-driven manner to representative normal patterns observed during training. For each selected memory item, a query-conditioned representation is computed through patch-level attention, and a gated update rule determines how much information from the current query is incorporated into the memory.

$$\mathbf{v}_{(k)} = \text{Attention}(\mathbf{m}_{(k)}, \mathbf{q}', \mathbf{q}'),$$

$$\boldsymbol{\psi}_{(k)} = \sigma\left(\mathbf{m}_{(k)}\mathbf{U}_\psi + \mathbf{v}_{(k)}\mathbf{W}_\psi\right),$$

$$\tilde{\mathbf{m}}_{(k)} = \left(1 - \boldsymbol{\psi}_{(k)}\right) \odot \mathbf{m}_{(k)} + \boldsymbol{\psi}_{(k)} \odot \mathbf{v}_{(k)}, \qquad k = 1, \ldots, K.$$

where $\mathbf{U}_\psi, \mathbf{W}_\psi \in \mathbb{R}^{d_{\text{model}} \times d_{\text{model}}}$ are learnable projection matrices, $\sigma(\cdot)$ denotes the element-wise sigmoid function.

**Query update**   In the query update stage, we refine $\mathbf{q}'$ into a memory-based representation $\tilde{\mathbf{q}}$ using patch-level attention over the selected memory items. This step incorporates retrieved normal memory patterns into the query representation before it is passed to the decoder. For each selected memory item $\mathbf{m}_{(k)}$, we compute an intermediate memory-based representation $\tilde{\mathbf{q}}_{(k)}$. The final refined representation $\tilde{\mathbf{q}}$ is obtained

by taking the weighted sum of the $K$ intermediate representations:

$$\tilde{\mathbf{q}}_{(k)} = \text{Attention}(\mathbf{q}', \mathbf{m}_{(k)}, \mathbf{m}_{(k)}), \qquad \tilde{\mathbf{q}} = \sum_{k=1}^{K} w_{(k)} \, \tilde{\mathbf{q}}_{(k)}.$$

where $w_{(k)}$ denotes the retrieval weight computed from the $k$-th selected score in the memory selection stage.

**Query alignment**  The resulting $\tilde{\mathbf{q}}$ is then combined with the original $\mathbf{q}$ in a patch-aligned manner, where each patch-level memory item contributes only to the corresponding patch position. This combined representation is then used as the input to the shared decoder. This ordered memory structure preserves the positional structure of the sequence and captures the underlying patterns in each patch.

**Memory dynamics**  During training, the memory is updated online at the sample level within each mini-batch. The memory values are not directly optimized by gradients, but are sequentially revised by the gated update rule according to the representations of previously processed samples and their order. The update rule is controlled by the projection matrices $\mathbf{U}_\psi$ and $\mathbf{W}_\psi$, which are learned from the reconstruction loss through standard mini-batch optimization. At inference time, the final trained memory is kept fixed and used for top-$K$ memory retrieval and memory-based query refinement, without test-time memory updates.

## 3.2  Multi-domain Training

Unlike the conventional one-model-per-domain paradigm, we train a single unified model that is jointly optimized on the entire dataset collection, enabling it to detect anomalies in time series with heterogeneous lengths and characteristics. During multi-domain training, the patch-based memory module is updated in a data-driven manner, allowing memory items to adapt to representative normal patterns from different domains. At initialization, the number of memory items is set equal to the number of user-defined domains, which provides a balanced starting point but does not enforce a strict one-to-one correspondence. Through data-driven updates, a single memory item may accumulate information from multiple domains and evolve into a domain-general feature. This training strategy facilitates knowledge sharing while maintaining a single model architecture in which the memory captures diverse normal patterns.

# 4  Experiments

**Datasets**  We evaluate our model on the TSB-AD-U benchmark (Liu & Paparrizos, 2024), consisting of 870 univariate time series across 23 datasets. The benchmark is constructed through an organized curation process: both univariate and multivariate datasets are collected, with multivariate data converted into univariate channels. Channels unrelated to anomaly labels, such as categorical, binary, or noisy channels, are discarded. Mislabeled or low-quality series are removed through algorithmic tests and human review, and balanced sampling is applied to ensure fairness. The collection also includes sequences from public benchmarks such as MSL (Mars Science Laboratory rover), SMD (Server Machine Dataset), and the UCR Anomaly Archive (Wu & Keogh, 2022). Further dataset details are provided in Appendix A.

**Experimental setup**  We follow the benchmark-provided train/test split for each time series in TSB-AD-U. The training segment corresponds to an initial subset of the full series, whereas the evaluation sequence consists of the full series, including the training segment. Normalization statistics are fitted using only the training segment of each series and then applied to the entire series. In the one-model-per-domain setting, we group the 350 Eval time series into 32 predefined domains based on dataset metadata and train one model per domain using only the training segments of the series assigned to that domain. The trained domain model is then evaluated on the evaluation sequences of the series in the same domain. We evaluate anomaly detection performance using threshold-independent metrics, such as the Area Under the Curve (AUC) and Volume Under the Surface (VUS) (Paparrizos et al., 2022). The reported benchmark-level scores are computed independently for each series and then averaged over the 350 Eval time series.

**Implementation details** For MOMEMTO and MOMENT, we generate fixed-length inputs by applying non-overlapping windows of length 512 to each normalized time series. The models are trained using the reconstruction loss $\mathcal{L}_{\text{rec}} = ||X - \hat{X}||_2^2$. During inference, the anomaly score at each time point is computed as the point-wise squared reconstruction error, $s_t = (x_t - \hat{x}_t)^2$. More detailed information on hyperparameter settings can be found in Appendix B.

## 4.1 Main Results

In this experiment, we evaluate the performance of MOMEMTO on time series anomaly detection by comparing it with 15 baseline methods. The baselines consist of three categories: classical approaches (OCSVM (Schölkopf et al., 1999), LOF (Breunig et al., 2000), Isolation Forest (Liu et al., 2008)), Deep learning models (LSTM-AD (Malhotra et al., 2015), Donut (Xu et al., 2018), OmniAnomaly (Su et al., 2019), TranAD (Tuli et al., 2022), Anomaly Transformer (Xu et al., 2022), MEMTO, TimesNet (Wu et al., 2023), KAN-AD (Zhou et al., 2025)), and pre-trained TSFMs (TimesFM (Das et al., 2024), Chronos (Ansari et al., 2024), MOMENT (Goswami et al., 2024), DADA (Shentu et al., 2025)).

Table 2: Model performance comparison on the benchmark. Metrics are reported as percentages (%). Subscripts indicate training settings: $md$ = multi-domain joint fine-tuning of a single model across all domains; models without a subscript denote the one-model-per-domain setting.

| Category | Model | AUC-PR | AUC-ROC | VUS-PR | VUS-ROC |
|---|---|---|---|---|---|
| Classical | LOF | $11.64_{\pm0.00}$ | $51.56_{\pm0.00}$ | $14.14_{\pm0.00}$ | $61.32_{\pm0.00}$ |
| | OCSVM | $32.60_{\pm0.00}$ | $68.62_{\pm0.00}$ | $33.03_{\pm0.00}$ | $75.34_{\pm0.00}$ |
| | Isolation Forest | $32.97_{\pm0.05}$ | $69.21_{\pm0.48}$ | $\underline{36.08}_{\pm0.10}$ | $75.95_{\pm0.24}$ |
| Deep learning | MEMTO | $6.93_{\pm1.20}$ | $50.70_{\pm1.20}$ | $10.51_{\pm1.06}$ | $54.33_{\pm0.73}$ |
| | Anomaly Transformer | $8.76_{\pm0.58}$ | $49.45_{\pm1.12}$ | $12.84_{\pm0.51}$ | $56.29_{\pm0.91}$ |
| | TimesNet | $16.22_{\pm0.65}$ | $59.75_{\pm0.54}$ | $22.82_{\pm0.90}$ | $71.08_{\pm0.20}$ |
| | TranAD | $21.70_{\pm0.31}$ | $59.00_{\pm0.35}$ | $25.46_{\pm0.22}$ | $69.09_{\pm0.27}$ |
| | Donut | $22.45_{\pm0.08}$ | $62.81_{\pm0.21}$ | $27.29_{\pm0.07}$ | $72.46_{\pm0.12}$ |
| | OmniAnomaly | $25.22_{\pm0.03}$ | $63.15_{\pm0.06}$ | $30.19_{\pm0.02}$ | $74.24_{\pm0.03}$ |
| | KAN-AD | $29.71_{\pm1.38}$ | $68.37_{\pm0.38}$ | $31.09_{\pm0.86}$ | $75.01_{\pm0.36}$ |
| | LSTM-AD | $33.64_{\pm0.33}$ | $71.97_{\pm0.66}$ | $33.35_{\pm0.43}$ | $77.96_{\pm0.58}$ |
| Foundation | Chronos | $27.02_{\pm0.16}$ | $65.78_{\pm0.02}$ | $27.30_{\pm0.26}$ | $72.70_{\pm0.03}$ |
| | Chronos$_{md}$ | $26.43_{\pm0.18}$ | $65.48_{\pm0.04}$ | $26.70_{\pm0.12}$ | $72.46_{\pm0.02}$ |
| | MOMENT | $30.22_{\pm0.00}$ | $70.04_{\pm0.01}$ | $30.55_{\pm0.01}$ | $76.55_{\pm0.01}$ |
| | MOMENT$_{md}$ | $29.58_{\pm0.03}$ | $69.33_{\pm0.05}$ | $29.89_{\pm0.04}$ | $76.82_{\pm0.00}$ |
| | TimesFM | $32.79_{\pm0.52}$ | $70.70_{\pm0.33}$ | $33.95_{\pm0.47}$ | $76.94_{\pm0.11}$ |
| | TimesFM$_{md}$ | $26.67_{\pm1.13}$ | $66.95_{\pm0.35}$ | $28.82_{\pm0.68}$ | $74.07_{\pm0.38}$ |
| | DADA | $32.20_{\pm0.30}$ | $\underline{72.86}_{\pm0.22}$ | $32.53_{\pm0.40}$ | $78.84_{\pm0.15}$ |
| | DADA$_{md}$ | $33.08_{\pm0.18}$ | $70.04_{\pm0.13}$ | $33.22_{\pm0.22}$ | $76.92_{\pm0.10}$ |
| | **MOMEMTO** | $\underline{34.87}_{\pm0.16}$ | $72.38_{\pm0.25}$ | $35.13_{\pm0.11}$ | $\underline{79.48}_{\pm0.19}$ |
| | **MOMEMTO**$_{md}$ | $\mathbf{36.35}_{\pm0.46}$ | $\mathbf{74.83}_{\pm0.15}$ | $\mathbf{37.62}_{\pm0.50}$ | $\mathbf{80.95}_{\pm0.25}$ |

Table 2 shows the evaluation results on the benchmark. Higher values in these metrics indicate more accurate anomaly detection. Among all compared models, **MOMEMTO$_{md}$**, as a single model, achieves the best performance across all evaluation criteria. Moreover, multi-domain training does not uniformly improve anomaly detection performance across TSFMs. Most TSFMs show little to no gain, whereas MOMEMTO benefits clearly from this setting. A direct comparison with its backbone TSFM, MOMENT, shows that MOMEMTO consistently surpasses its backbone across metrics and settings. This demonstrates that the patch-based memory module effectively mitigates over-generalization and improves detection capability, even when built upon the same pre-trained encoder, highlighting the importance of architectural design tailored to anomaly detection.

**Few-shot learning** To evaluate the robustness of our approach under limited data scenarios, we conduct a few-shot learning experiment. In this setting, the ratio of training data is gradually varied from 10% to 80%, and the performance of MOMEMTO is compared with MOMENT under the condition that only a small fraction of samples is available for fine-tuning.

Figure 2 shows the effect of training data ratio on AUC-PR. Across all ratios, MOMEMTO consistently outperforms MOMENT, highlighting the advantage of the patch-based memory module. The gap is substantial in the few-shot regime (10%–30%), where both the one-model-per-domain and multi-domain variants achieve noticeably higher scores than the corresponding MOMENT variants. As the proportion of training data increases, MOMEMTO continues to improve in both settings, while MOMENT shows marginal or no further improvement. Moreover, MOMEMTO$_{\mathrm{md}}$ achieves consistently higher performance than MOMEMTO without multi-domain training, indicating that multi-domain training further strengthens the memory module by improving data efficiency. Overall, these results demonstrate that MOMEMTO is robust under limited data and benefits even more from additional data when trained in the multi-domain setting.

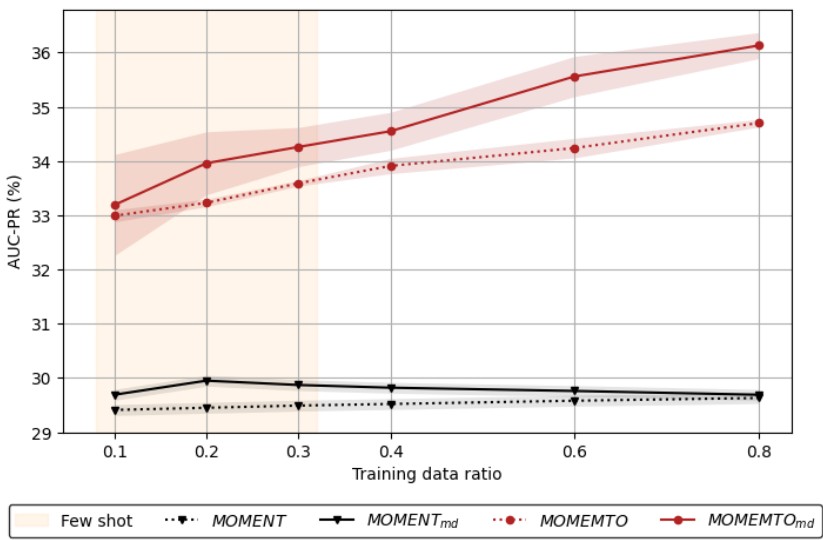

Figure 2: Effect of the training data ratio on AUC-PR.

## 4.2 Model Analysis

**Ablation study** To investigate the contribution of each component in MOMEMTO, we conduct an ablation study by varying the encoder initialization (scratch vs. pre-trained) and the use of the patch-based memory module.

As shown in Table 3, both components contribute significantly to performance improvement. Replacing a scratch encoder with a pre-trained one improves AUC-PR ($22.44 \rightarrow 29.58$), while adding the memory module to a scratch encoder yields an even larger gain ($22.44 \rightarrow 33.63$). The combination of both components achieves the best performance, reaching an AUC-PR of 36.35. Similar trends are consistently observed across other metrics. These results indicate that the two components provide complementary benefits: the patch-based memory module mitigates over-generalization, and the pre-trained encoder provides a strong representational foundation that amplifies the memory module's effectiveness. Additional ablation results on the decoder input representation are provided in Appendix C.5.

**Comparison of anomaly score distributions** Figure 3 presents the distributions of anomaly scores for normal and anomalous samples across a subset of domains. Each boxplot shows the distributions of normal and anomalous samples within a domain, enabling direct comparison of the models' discriminative ability. Additional results for the remaining domains are provided in Appendix C.11. Overall, MOMENT exhibits limited separation between normal and anomalous samples, with substantial overlap in their score

Table 3: Ablation results of pre-trained encoder and patch-based memory module. 'PMM' denotes the patch-based memory module.

| Encoder | PMM | AUC-PR | AUC-ROC | VUS-PR | VUS-ROC |
|---------|-----|--------|---------|--------|---------|
| scratch | × | 22.44 | 67.35 | 18.90 | 66.50 |
| scratch | ○ | 33.63 | 72.17 | 34.65 | 78.78 |
| pre-trained | × | 29.58 | 69.33 | 29.89 | 76.82 |
| pre-trained | ○ | **36.35** | **74.83** | **37.62** | **80.95** |

distributions. In contrast, MOMEMTO often yields a clearer margin between the two groups: anomaly scores for anomalous samples tend to shift toward higher values, whereas those of normal samples remain clustered around low values. This tendency suggests that MOMEMTO improves the distinction between normal and anomalous patterns in the majority of domains.

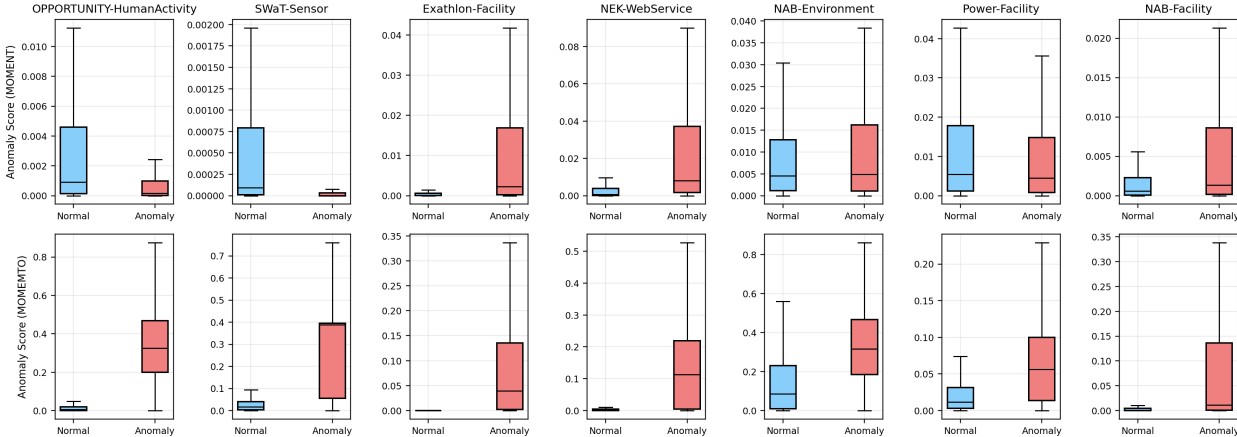

Figure 3: Comparison of anomaly score distributions between normal (blue) and anomalous (red) samples across a subset of domains. The upper row shows the results obtained with the backbone model MOMENT, while the lower row corresponds to our proposed model.

**Over-generalization alleviation** As shown in Figure 4, the pre-trained MOMENT backbone exhibits strong zero-shot reconstruction capability across diverse time-series patterns. Even without task-specific fine-tuning, MOMENT accurately reconstructs a variety of simple temporal structures, including sinusoid, random walk and shifted variants of similar patterns. In particular, when the test sequences share a similar temporal shape with the training data but differ in their mean level, MOMENT often reconstructs these shifted sequences with low reconstruction error.

Figure 5 illustrates this behavior under controlled mean-level shifts, showing training samples, test samples, and window-level reconstruction errors across mean levels. The dashed line represents the threshold defined by the maximum training reconstruction error. In the simple level-shift setting, MOMENT assigns low reconstruction errors to several out-of-range shifted sequences, suggesting that the reconstruction is driven more by temporal shape similarity than by the normal mean-level range. In the noisy setting, reconstruction errors of normal samples also increase, further obscuring the level difference between normal and shifted inputs. This indicates that MOMENT's over-generalization can make it insensitive to mean-level deviations, both for simple patterns and for noisier shifted patterns.

In contrast, MOMEMTO reconstructs shifted inputs closer to the learned normal range rather than freely extrapolating to unseen mean levels. As a result, it tends to produce larger reconstruction errors for mean-

shift anomalies and yields clearer separation between normal and abnormal sequences. This tendency is further supported by the success and failure case analysis on the benchmark time series in Appendix C.1.

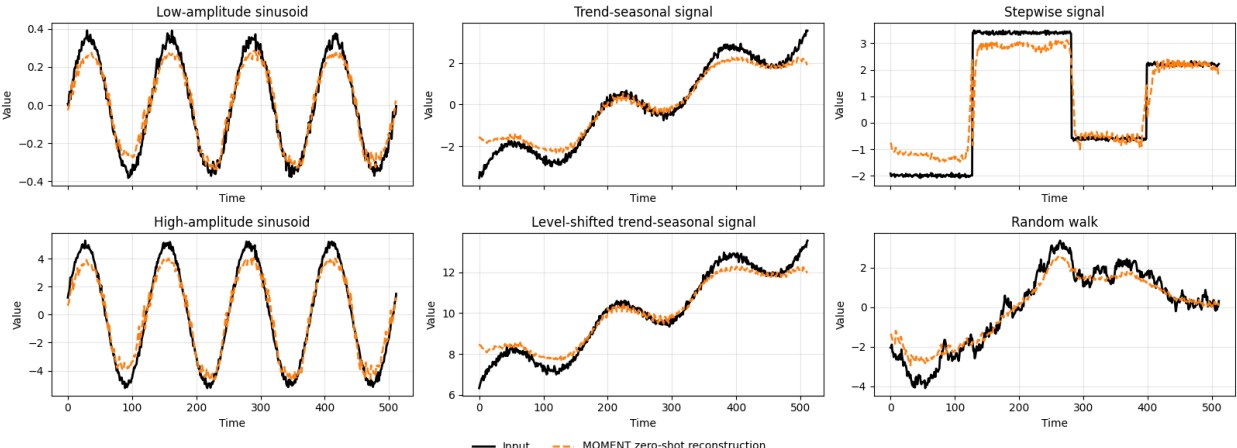

Figure 4: Zero-shot reconstruction examples of the pre-trained MOMENT backbone. MOMENT captures diverse synthetic patterns, including sinusoidal, trend-seasonal, stepwise, and stochastic signals, without task-specific fine-tuning.

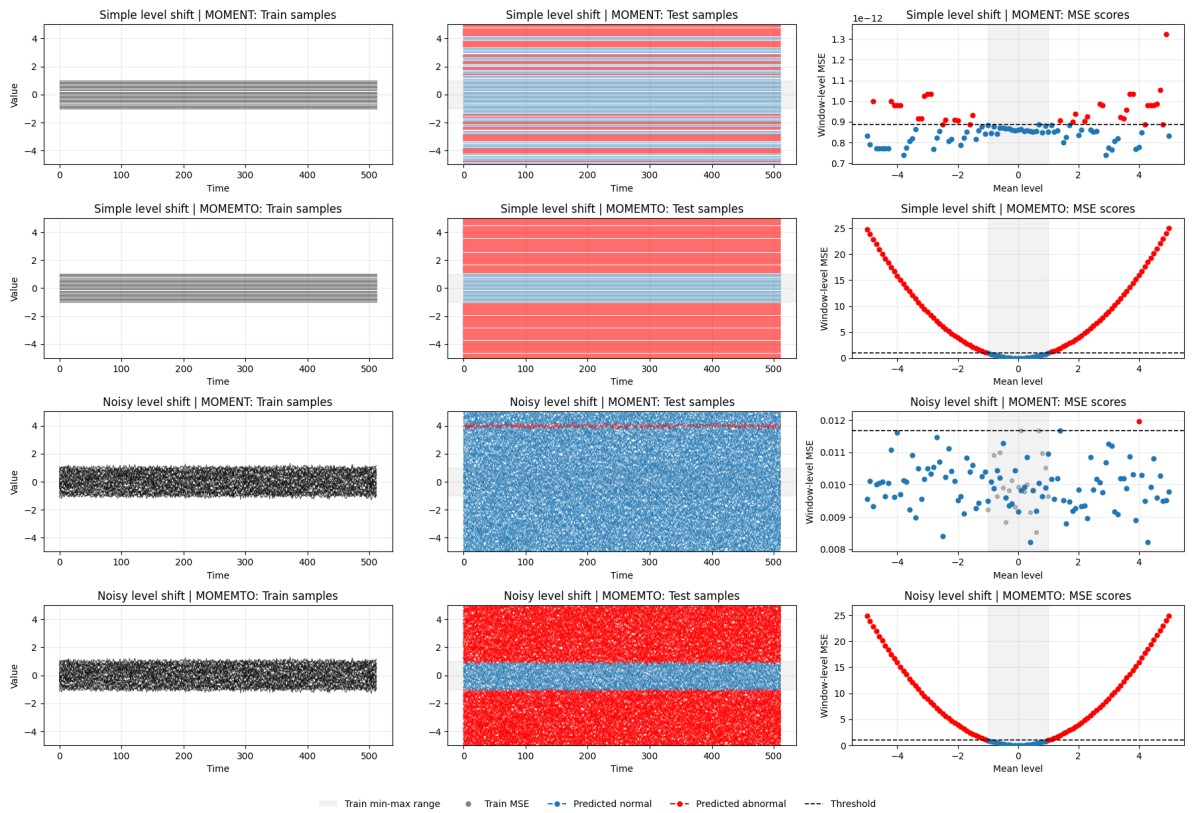

Figure 5: Controlled mean-level shift analysis under simple and noisy settings. The left/middle columns show training/test samples, and the right column shows reconstruction errors across mean levels. The dashed line is the maximum training-error threshold, and blue/red indicates predicted normal/abnormal.

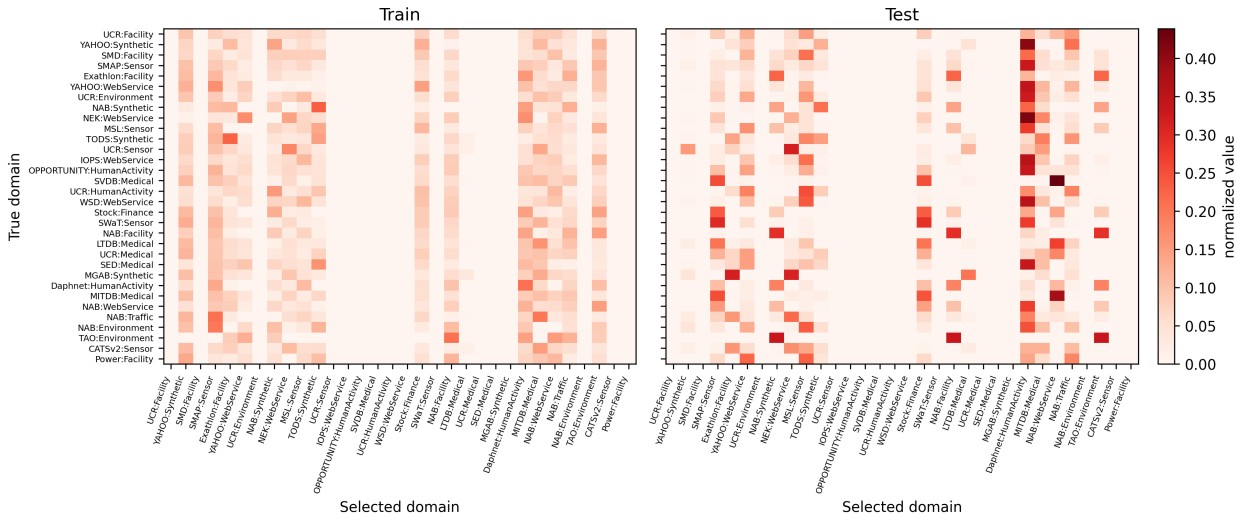

Figure 6: Visualization of row-normalized accumulated domain-to-memory similarity over the entire training process (left) and at test time (right). Each heatmap shows how subsequences from a true domain reference memory items across domains.

**Data-driven memory utilization**  We visualize in Figure 6 how memory items are accessed for subsequences from different domains. Each heatmap displays the accumulated similarity between the true domain of the input and the selected memory domains, normalized such that each row sums to one. The left illustrates updates during the entire training, and the right shows references by new inputs at test time. With the data-driven strategy, no domain labels are provided during training or testing. Updates tend to concentrate on a subset of memory items, rather than being evenly distributed across domains, and these items are consistently referenced at test time.

To validate the effectiveness of our strategy, we compare it with two alternative strategies: (i) freezing the memory items without updates and (ii) updating only the memory item initially assigned to the input's domain (own-domain update). Both baselines use the memory item corresponding to the input's domain during training and testing. Table 4 summarizes the performance comparison. Even though the initial allocation of memory items to domains does not remain aligned during training, our data-driven strategy still achieves higher performance while utilizing fewer memory items than alternative strategies. This indicates that the memory module has learned a stable set of domain-general features, which are effectively reused across unseen inputs. This concentration behavior is further analyzed in Appendix C.4, where we vary the number of memory items and observe similar post-training concentration patterns across different memory capacities.

Table 4: Performance comparison of different memory update strategies. *No update*: memory frozen after initialization; *Own-domain update*: update only the item initially assigned to the input's domain; *Data-driven update (ours)*: update the similarity-retrieved item regardless of initial domain. At test time, both baselines use the memory item corresponding to the input's domain.

| Strategy | AUC-PR | AUC-ROC | VUS-PR | VUS-ROC |
|---|---|---|---|---|
| No update | 34.97 | 73.77 | 36.91 | 80.19 |
| Own-domain update | 35.16 | 73.64 | 36.60 | 80.06 |
| Data-driven update (ours) | **36.35** | **74.83** | **37.62** | **80.95** |

**Compatibility with other TSFMs**  We further evaluate the effectiveness of the patch-based memory module on other patch-based and reconstruction-based TSFMs. We incorporate PMM into UniTS (Gao et al., 2024) and UP2ME (Zhang et al., 2024) and compare the resulting models with their original counterparts. As shown in Table 5, PMM consistently improves both AUC-PR and AUC-ROC across all evaluated backbones. MOMENT combined with PMM achieves the strongest overall performance, which motivates our choice of MOMENT as the main backbone.

Table 5: Effectiveness of the patch-based memory module (PMM) across different TSFM backbones. PMM is incorporated into patch-based and reconstruction-based TSFMs, and the resulting performance is compared with that of the original models.

| Model | Original TSFM | | TSFM + PMM | |
|---|---|---|---|---|
| | AUC-PR | AUC-ROC | AUC-PR | AUC-ROC |
| UniTS$_{md}$ | 21.96 | 67.10 | **33.99** | **72.24** |
| UP2ME$_{md}$ | 26.67 | 67.51 | **33.96** | **72.03** |
| MOMENT$_{md}$ | 29.58 | 69.33 | **36.35** | **74.83** |

## 5 Conclusion

We introduce MOMEMTO, a time series foundation model for anomaly detection enhanced with a patch-based memory module. By leveraging a pre-trained encoder and the patch-based memory module, our approach mitigates over-generalization and achieves higher AUC and VUS metrics than the evaluated baselines across 23 univariate benchmark datasets, with notable gains in few-shot and multi-domain settings. While MOMEMTO shows promising results, this study focuses on univariate time series anomaly detection, and several important aspects remain unexplored. Extending MOMEMTO to multivariate settings would require additional design choices, such as modeling cross-channel dependencies and organizing memory across variables. We leave these directions for future work to broaden the applicability of our approach.

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

## A  TSB-AD-U Benchmark

In this section, we provide details of the TSB-AD-U benchmark (Liu & Paparrizos, 2024) used in our experiments. The benchmark comprises 23 univariate time series datasets with a total of 870 time series, collected from diverse domains. Table 6 summarizes the dataset composition, including the number of series, average sequence length, anomaly counts, and anomaly ratios. The benchmark is divided into two partitions: the Eval set, which contains 350 time series and is designated for evaluation, and the Tuning set, which contains 48 time series and is used for hyperparameter optimization. Table 7 summarizes the statistics of the Eval and Tuning sets.

Table 6: Summary characteristics of 23 datasets included in TSB-AD-U. '-' in the 2nd column indicates this dataset is transformed from the multivariate dataset. The 'Category' column indicates whether the datasets feature point anomalies (P) or sequence anomalies (Seq).

| Name | # TS Collected | # TS Curated | Avg Dim | Avg TS Len | Avg # Anomaly | Avg Anomaly Len | Anomaly Ratio | Category |
|------|---------------|--------------|---------|-----------|---------------|-----------------|---------------|----------|
| UCR | 250 | 228 | 1 | 67818.7 | 1 | 198.9 | 0.6% | P&Seq |
| NAB | 58 | 28 | 1 | 5099.7 | 1.6 | 370.1 | 10.6% | Seq |
| YAHOO | 367 | 259 | 1 | 1560.2 | 5.5 | 2.5 | 0.6% | P&Seq |
| IOPS | 58 | 17 | 1 | 72792.3 | 25.6 | 48.7 | 1.3% | Seq |
| MGAB | 10 | 9 | 1 | 97777.8 | 9.7 | 20.0 | 0.2% | Seq |
| WSD | 210 | 111 | 1 | 17444.5 | 5.1 | 25.4 | 0.6% | Seq |
| SED | 6 | 3 | 1 | 23332.3 | 14.7 | 64.0 | 4.1% | Seq |
| TODS | 15 | 15 | 1 | 5000.0 | 97.3 | 18.7 | 6.3% | P&Seq |
| NEK | 48 | 9 | 1 | 1073.0 | 2.9 | 51.1 | 8.0% | P&Seq |
| Stock | 90 | 20 | 1 | 15000.0 | 1246.9 | 1.1 | 9.4% | P&Seq |
| Power | 1 | 1 | 1 | 35040.0 | 4 | 750 | 8.5% | Seq |
| Daphnet (U) | - | 1 | 1 | 38774.0 | 6 | 384.3 | 5.9% | Seq |
| CATSv2 (U) | - | 1 | 1 | 300000.0 | 19.0 | 778.9 | 4.9% | Seq |
| SWaT (U) | - | 1 | 1 | 419919.0 | 27.0 | 1876.0 | 12.1% | Seq |
| LTDB (U) | - | 9 | 1 | 99700.0 | 127.5 | 144.5 | 18.6% | Seq |
| TAO (U) | - | 3 | 1 | 10000.0 | 838.7 | 1.1 | 9.4% | P&Seq |
| Exathlon (U) | - | 32 | 1 | 44075.8 | 3.1 | 1577.3 | 11.0% | Seq |
| MITDB (U) | - | 8 | 1 | 631250.0 | 68.7 | 451.9 | 4.2% | Seq |
| MSL (U) | - | 9 | 1 | 3492.0 | 1.3 | 130.0 | 5.8% | Seq |
| SMAP (U) | - | 19 | 1 | 7700.2 | 1.2 | 210.1 | 2.8% | Seq |
| SMD (U) | - | 38 | 1 | 24207.7 | 2.4 | 173.7 | 2.0% | Seq |
| SVDB (U) | - | 20 | 1 | 171380.0 | 36.4 | 292.5 | 3.6% | Seq |
| OPP (U) | - | 29 | 1 | 16544.8 | 1.4 | 653.4 | 6.4% | Seq |

Table 7: Statistics of the TSB-AD-U benchmark splits.

| | Split | # TS | Avg Length | Avg Anomaly Length | Avg # Anomalies | Anomaly Ratio |
|--|-------|------|------------|--------------------|-----------------|---------------|
| | All | 870 | 38814.1 | 179.5 | 39.7 | 2.4% |
| TSB-AD-U | Eval | 350 | 51886.7 | 321.3 | 46.6 | 4.5% |
| | Tuning | 48 | 47143.3 | 185.9 | 82.6 | 3.5% |

## B  Implementation Details

All experiments in this paper are conducted on the Eval set of the TSB-AD-U benchmark, which contains 350 univariate time series. We describe below how the domain is defined, the training settings we consider, and the training configurations we adopt in our experiments.

### B.1  Domain Partition

We assign a domain label to each time series in the Eval set. A domain label is defined as a tuple (`Dataset`, `Sub-domain`), where the Dataset indicates the source collection and the Sub-domain specifies the actual

domain of the time series. Time series characteristics may vary across domains within the same dataset, and they may also differ across datasets even when they share the same domain. To capture this heterogeneity, domain labels serve as a heuristic for balanced memory initialization.

Each file in the benchmark follows a structured naming convention.

For example:

`001_NAB_id_1_Facility_tr_1007_1st_2014.csv`

In this case:

- Dataset:= `NAB`

- Sub-domain:= `Facility`

- **Domain label**:= `(NAB, Facility)`

In total, the 350 univariate time series in the Eval set are grouped into 32 domains according to this labeling. These domain labels are used only during the memory initialization stage.

## B.2 Training Settings

We distinguish between two training settings used in our experiments:

- **One-model-per-domain:** A separate model is trained for each domain defined in Appendix B.1. Since the 350 time series in the Eval set are grouped into 32 domains, this setting results in 32 independently trained models.

- **Multi-domain training:** In contrast, our proposed approach jointly optimizes a single model across all domains in the Eval set.

## B.3 Baselines and Hyperparameter Settings

All baseline models are trained on the training segment and evaluated on the test sequence as defined in the benchmark. For fair comparison, all methods follow the same benchmark split and use the benchmark-provided tuning set for hyperparameter selection. We report the best configuration selected through this tuning procedure, including the window size.

For non-TSFM baselines, classical methods are fitted using the selected hyperparameters, while neural models are trained from scratch; when applicable, early stopping is performed using a held-out validation subset split from the training segments. TSFM baselines are loaded from their pre-trained checkpoints and fine-tuned on the training segments. Their fine-tuning is conducted with 2 epochs and a batch size of 32. A subset of models, such as MOMENT, Anomaly Transformer, MEMTO, and DADA, use non-overlapping windows to respect the original implementations in their respective papers. The remaining models segment each time series into fixed-length sliding windows with stride 1.

For MOMEMTO, we use MOMENT-large (Goswami et al., 2024) as the backbone model, and most of the hyperparameters remain consistent with the original MOMENT configuration. Only the parameters related to the patch-based memory module are newly introduced. Table 9 summarizes the hyperparameter settings and environment used in our experiments.

Table 8: Hyperparameter settings for baseline models.

| Model | Hyperparameter |
|---|---|
| LOF | n_neighbors = 50, metric = minkowski |
| OCSVM | periodicity = 2, kernel = rbf |
| Isolation Forest | n_estimators = 200 |
| MEMTO | win_size = 100, lr = 0.00005, $\lambda$ = 0.01, n_memory = 10 |
| Donut | win_size = 60, lr = 0.0001 |
| TimesNet | win_size = 32, lr = 0.0001 |
| Anomaly Transformer | win_size = 100, lr = 0.001 |
| TranAD | win_size = 10, lr = 0.0001 |
| OmniAnomaly | win_size = 5, lr = 0.002 |
| LSTM-AD | win_size = 100, lr = 0.0008 |
| Chronos | model = base, win_size = 100, lr = 0.0001 |
| TimesFM | win_size = 96, lr = 0.0001 |
| DADA | win_size = 100, lr = 0.0001 |
| MOMENT | win_size = 512, lr = 0.0001 |
| UniTS | win_size = 512, lr = 0.0001 |
| UP2ME | win_size = 500, lr = 0.0001 |

Table 9: Hyperparameter settings for MOMEMTO and experimental environment.

| Category | Setting |
|---|---|
| Window size | 512 |
| Patch length ($L$) | 8 |
| Patch stride length | 8 |
| Number of patches ($N$) | 64 |
| $d_{model}$ | 1024 |
| Number of referenced items ($K$) | 3 |
| Temperature parameter ($\tau$) | 0.3 |
| Memory initialization sampling ratio per domain | 0.1 |
| Learning rate | 0.0001 |
| Epochs | 2 |
| Optimizer | Adam |
| Loss function | Mean Squared Error (MSE) |
| Anomaly Score | Pointwise squared reconstruction error; $s_t = (\hat{x}_t - x_t)^2$ |
| GPU | NVIDIA GeForce RTX 4090 (24GB) |
| Framework | PyTorch 2.7.0 |

### B.4 Algorithms

Algorithms 1 and 2 illustrate the overall mechanism of our model. They present the matrix operation version of the forward process when a single input subsequence is given.

---

**Algorithm 1** Proposed Method **MOMEMTO**

---

**Input** $\mathbf{x} \in \mathbb{R}^{N \times L}$ : input time series, $[\texttt{mask}] = \{0,1\}^N$, $\mathcal{M} \in \mathbb{R}^{M \times N \times d_{model}}$: domain-specific initialized memory items

**Training params** $f_e$: encoder, $f_d$: decoder, $f_m$: patch-based memory module

1: $\mathbf{q} \leftarrow f_e(\mathbf{x}, [\texttt{mask}])$             $\triangleright \ \mathbf{q} \in \mathbb{R}^{P \times d_{model}}$

2: $\mathbf{q}' \leftarrow \text{L2Norm}(\mathbf{q}, dim = 1)$

3: $\tilde{\mathbf{q}} \leftarrow f_m(\mathbf{q}', \mathcal{M}, [\texttt{mask}])$

4: $\hat{\mathbf{q}} \leftarrow concat([\mathbf{q}, \tilde{\mathbf{q}}], dim = 1)$          $\triangleright \ \hat{\mathbf{q}} \in \mathbb{R}^{P \times (2 \cdot d_{model})}$

5: $\hat{\mathbf{x}} \leftarrow f_d(\hat{\mathbf{q}})$

6: **return** $\hat{\mathbf{x}}$              $\triangleright$ Reconstructed time series

---

**Algorithm 2** Proposed **Patch-based memory module**

---

**Require:** $\mathbf{q}' \in \mathbb{R}^{P \times d_{\text{model}}}$: normalized queries, $[\texttt{mask}] = \{0,1\}^N$, $\mathcal{M} \in \mathbb{R}^{M \times N \times d_{\text{model}}}$: memory items

**Require:** $\mathbf{U}_\psi, \mathbf{W}_\psi \in \mathbb{R}^{d_{\text{model}} \times d_{\text{model}}}$: linear projection matrices

1: $\mathcal{M} \in \mathbb{R}^{M \times P \times d_{\text{model}}} \xleftarrow{[\texttt{mask}]} \mathcal{M} \in \mathbb{R}^{M \times N \times d_{\text{model}}}$     $\triangleright$ Memory alignment

2: Reshape: $\mathbf{q}' \in \mathbb{R}^{1 \times (P \cdot d_{\text{model}})}, \quad \mathcal{M} \in \mathbb{R}^{M \times (P \cdot d_{\text{model}})}$

3: $\mathbf{s} \leftarrow \mathbf{q}' \mathcal{M}^\top / \tau$           $\triangleright$ Sequence-level similarity scores

4: $(\mathbf{s}_\mathcal{I}, \mathcal{I}) \leftarrow \text{TopK}(\mathbf{s}, K)$       $\triangleright$ Select top-$K$ memory items from raw scores

5: $\boldsymbol{\lambda} \leftarrow \text{softmax}(\mathbf{s}_\mathcal{I})$        $\triangleright$ Normalize only over selected top-$K$ scores

6: Reshape: $\mathbf{q}' \in \mathbb{R}^{P \times d_{\text{model}}}, \quad \mathcal{M} \in \mathbb{R}^{M \times P \times d_{\text{model}}}$

7: $\tilde{\mathbf{q}} \leftarrow \mathbf{0}_{P \times d_{\text{model}}}$

8: **for** $r \leftarrow 1$ **to** $K$ **do**

9:   $j \leftarrow \mathcal{I}_r, \quad \mathbf{m}_j \leftarrow \mathcal{M}_j$         $\triangleright \ \mathbf{m}_j \in \mathbb{R}^{P \times d_{\text{model}}}$

10:   $\mathbf{A}_v \leftarrow \text{softmax}(\mathbf{m}_j \mathbf{q}'^\top / \sqrt{d_{\text{model}}})$    $\triangleright$ Attention weights for memory update

11:   $\mathbf{v}_j \leftarrow \mathbf{A}_v \mathbf{q}'$          $\triangleright \ \mathbf{v}_j \in \mathbb{R}^{P \times d_{\text{model}}}$

12:   $\boldsymbol{\psi}_j \leftarrow \text{sigmoid}(\mathbf{m}_j \mathbf{U}_\psi + \mathbf{v}_j \mathbf{W}_\psi)$     $\triangleright \ \boldsymbol{\psi}_j \in \mathbb{R}^{P \times d_{\text{model}}}$

13:   $\tilde{\mathbf{m}}_j \leftarrow (1 - \boldsymbol{\psi}_j) \odot \mathbf{m}_j + \boldsymbol{\psi}_j \odot \mathbf{v}_j$      $\triangleright$ Memory update

14:   **if** training **then**

15:    $\mathcal{M}_j \leftarrow \tilde{\mathbf{m}}_j$        $\triangleright$ Persist the selected memory update

16:    $\mathbf{m}_j \leftarrow \tilde{\mathbf{m}}_j$      $\triangleright$ Use the updated memory item for query update

17:   **end if**

18:   $\mathbf{A}_q \leftarrow \text{softmax}(\mathbf{q}' \mathbf{m}_j^\top / \sqrt{d_{\text{model}}})$    $\triangleright$ Attention weights for query update

19:   $\tilde{\mathbf{q}}_j \leftarrow \mathbf{A}_q \mathbf{m}_j$         $\triangleright \ \tilde{\mathbf{q}}_j \in \mathbb{R}^{P \times d_{\text{model}}}$

20:   $\tilde{\mathbf{q}} \leftarrow \tilde{\mathbf{q}} + \lambda_r \tilde{\mathbf{q}}_j$

21: **end for**

22: **return** $\tilde{\mathbf{q}}$           $\triangleright \ \tilde{\mathbf{q}} \in \mathbb{R}^{P \times d_{\text{model}}}$

---

## C   Additional Experimental Results

Unless otherwise specified, we refer to MOMENT$_{md}$ and MOMEMTO$_{md}$ as MOMENT and MOMEMTO, respectively, throughout Appendix C for simplicity.

Table 10: AUC-PR results by domain.

| Domain | LOF | OCSVM | IForest | MEMTO | Donut | TimesNet | A.T. | TranAD | OmniAnomaly | LSTM-AD | Chronos | TimesFM | MOMENT | MOMEMTO | MOMEMTO$_{md}$ |
|---|---|---|---|---|---|---|---|---|---|---|---|---|---|---|---|
| (CATSv2, Sensor) | 10.79 | 10.94 | 17.10 | 5.18 | 10.35 | 16.47 | 5.52 | 9.78 | 10.93 | 35.49 | 10.97 | 36.56 | 25.67 | 12.58 | 38.26 |
| (Daphnet, HumanActivity) | 5.32 | 20.03 | 31.35 | 6.38 | 18.91 | 36.23 | 7.09 | 19.86 | 19.89 | 35.87 | 31.30 | 37.30 | 33.25 | 20.14 | 10.95 |
| (Exathlon, Facility) | 22.64 | 73.76 | 97.57 | 12.09 | 77.84 | 47.87 | 13.43 | 56.38 | 74.36 | 60.30 | 45.23 | 56.64 | 52.33 | 89.88 | 90.37 |
| (IOPS, WebService) | 3.30 | 30.23 | 21.43 | 3.79 | 13.38 | 8.10 | 4.19 | 25.02 | 24.57 | 30.26 | 24.40 | 21.02 | 24.29 | 29.38 | 27.28 |
| (LTDB, Medical) | 17.85 | 26.51 | 55.15 | 19.57 | 25.30 | 26.24 | 23.68 | 25.06 | 26.08 | 21.40 | 21.14 | 24.60 | 23.59 | 26.67 | 29.92 |
| (MGAB, Synthetic) | 0.20 | 0.25 | 0.21 | 0.21 | 0.19 | 0.28 | 0.22 | 0.33 | 0.26 | 12.35 | 0.21 | 5.26 | 0.38 | 0.33 | 0.62 |
| (MITDB, Medical) | 4.48 | 7.89 | 18.36 | 4.87 | 7.81 | 6.43 | 7.07 | 8.11 | 7.92 | 6.70 | 4.72 | 8.43 | 6.31 | 10.05 | 15.09 |
| (MSL, Sensor) | 16.50 | 21.00 | 36.49 | 8.44 | 24.84 | 25.70 | 10.45 | 21.70 | 21.16 | 26.65 | 13.51 | 28.15 | 24.79 | 21.27 | 24.76 |
| (NAB, Environment) | 26.34 | 32.07 | 49.23 | 10.44 | 31.56 | 12.41 | 10.41 | 24.78 | 32.41 | 21.05 | 10.00 | 11.70 | 13.05 | 36.84 | 36.79 |
| (NAB, Facility) | 12.16 | 20.53 | 34.62 | 10.69 | 23.46 | 18.68 | 11.81 | 19.83 | 20.61 | 22.56 | 16.27 | 22.51 | 23.07 | 21.31 | 25.55 |
| (NAB, Synthetic) | 22.28 | 16.76 | 42.53 | 12.55 | 21.50 | 19.48 | 14.29 | 19.92 | 17.07 | 20.81 | 18.24 | 21.09 | 20.51 | 17.30 | 22.12 |
| (NAB, Traffic) | 10.36 | 14.40 | 28.19 | 10.27 | 15.34 | 13.89 | 10.67 | 15.75 | 14.66 | 12.99 | 13.08 | 15.26 | 14.31 | 14.40 | 14.42 |
| (NAB, WebService) | 12.18 | 13.50 | 13.37 | 12.02 | 13.60 | 18.56 | 12.78 | 13.60 | 13.69 | 13.88 | 14.38 | 17.35 | 16.39 | 13.66 | 19.97 |
| (NEK, WebService) | 19.26 | 69.14 | 67.97 | 11.27 | 40.59 | 29.25 | 13.16 | 63.33 | 68.77 | 44.57 | 36.19 | 36.69 | 37.16 | 69.94 | 64.03 |
| (OPPORTUNITY, HumanActivity) | 4.96 | 81.94 | 92.50 | 9.39 | 69.98 | 4.99 | 35.68 | 60.01 | 81.05 | 10.71 | 5.49 | 5.22 | 4.80 | 75.45 | 77.17 |
| (Power, Facility) | 8.64 | 9.13 | 7.06 | 8.64 | 9.24 | 7.35 | 8.90 | 11.07 | 9.06 | 7.87 | 7.89 | 7.81 | 7.94 | 9.90 | 21.02 |
| (SED, Medical) | 4.04 | 4.15 | 2.18 | 4.23 | 3.72 | 2.59 | 4.72 | 2.40 | 2.76 | 4.08 | 3.63 | 6.06 | 2.92 | 3.16 | 4.57 |
| (SMAP, Sensor) | 8.86 | 17.63 | 38.44 | 3.84 | 31.54 | 37.80 | 6.18 | 20.36 | 15.69 | 27.81 | 15.05 | 28.67 | 34.49 | 20.88 | 24.87 |
| (SMD, Facility) | 3.49 | 38.23 | 39.70 | 3.27 | 28.32 | 45.51 | 9.03 | 36.98 | 37.94 | 49.69 | 38.85 | 51.83 | 48.72 | 41.41 | 37.80 |
| (SVDB, Medical) | 3.73 | 7.57 | 34.44 | 4.01 | 7.37 | 7.35 | 4.29 | 7.76 | 7.74 | 8.24 | 4.81 | 8.29 | 6.81 | 15.14 | 16.10 |
| (SWaT, Sensor) | 9.56 | 74.84 | 74.71 | 16.05 | 75.07 | 10.87 | 34.21 | 74.91 | 75.35 | 73.98 | 42.79 | 20.69 | 7.88 | 75.62 | 75.07 |
| (Stock, Finance) | 46.80 | 99.71 | 10.11 | 20.09 | 8.71 | 8.84 | 8.25 | 8.01 | 8.32 | 98.96 | 94.52 | 97.07 | 98.33 | 98.13 | 58.51 |
| (TAO, Environment) | 100.00 | 99.94 | 12.01 | 15.41 | 12.25 | 13.46 | 11.70 | 11.49 | 11.70 | 99.26 | 55.54 | 95.46 | 99.92 | 99.94 | 73.21 |
| (TODS, Synthetic) | 15.03 | 22.39 | 7.07 | 7.82 | 7.00 | 9.33 | 8.24 | 6.25 | 6.12 | 42.01 | 36.79 | 54.06 | 42.52 | 21.76 | 34.15 |
| (UCR, Environment) | 0.66 | 1.17 | 0.91 | 0.59 | 0.81 | 2.23 | 1.49 | 0.89 | 1.16 | 21.81 | 17.82 | 18.23 | 6.74 | 2.58 | 2.61 |
| (UCR, Facility) | 0.43 | 1.60 | 9.91 | 0.40 | 1.91 | 1.10 | 0.81 | 2.27 | 1.97 | 1.65 | 0.75 | 1.11 | 1.67 | 2.48 | 4.24 |
| (UCR, HumanActivity) | 1.01 | 2.52 | 7.21 | 0.63 | 1.77 | 2.17 | 0.75 | 2.60 | 1.99 | 14.80 | 8.40 | 7.70 | 2.11 | 5.48 | 10.91 |
| (UCR, Medical) | 1.36 | 1.55 | 5.55 | 0.99 | 1.39 | 3.62 | 1.65 | 1.40 | 1.57 | 11.77 | 9.73 | 16.38 | 2.65 | 1.52 | 4.17 |
| (UCR, Sensor) | 1.01 | 2.97 | 8.16 | 1.11 | 2.87 | 2.92 | 1.36 | 3.34 | 3.06 | 6.92 | 2.80 | 10.84 | 6.96 | 2.99 | 4.92 |
| (WSD, WebService) | 0.92 | 22.61 | 4.98 | 6.20 | 3.87 | 4.05 | 3.02 | 13.23 | 17.95 | 39.37 | 32.44 | 32.60 | 28.26 | 24.28 | 21.32 |
| (YAHOO, Synthetic) | 63.54 | 65.71 | 0.18 | 24.92 | 0.18 | 0.93 | 0.16 | 0.20 | 1.57 | 99.74 | 85.49 | 93.94 | 99.79 | 66.17 | 80.67 |
| (YAHOO, WebService) | 22.34 | 35.46 | 13.50 | 4.14 | 6.11 | 4.76 | 2.81 | 9.76 | 11.86 | 64.01 | 73.92 | 79.01 | 75.85 | 39.37 | 48.91 |

## C.1  Qualitative Case Analysis of MOMENT and MOMEMTO

Table 10 summarizes more detailed AUC-PR results by domain, aggregated from 350 time series across 32 domains. We further conduct a qualitative case study comparing the raw input, reconstructed outputs, and anomaly scores of MOMENT and MOMEMTO at the individual time-series level.

For each test series, we compute the AUC-PR scores of MOMENT and MOMEMTO and categorize the cases into four groups: (i) both models succeed, (ii) MOMENT fails while MOMEMTO succeeds, (iii) MOMENT succeeds while MOMEMTO fails, and (iv) both models fail. From each category, we select a representative example that illustrates the typical qualitative pattern observed in that category. This categorization separates general detection behavior from cases where the memory module provides clear benefits or where the qualitative results reveal limitations related to annotation ambiguity or train–test distribution shift.

Figures 7–11 follow the same layout. The first row shows the full input series and the normalized anomaly-score profiles of both models. The blue region indicates the training segment, the red region marks the ground-truth anomaly, and the gray region denotes the interval zoomed in the following rows. The second row compares the zoomed input with the reconstructions generated by MOMENT and MOMEMTO$_{md}$. The third row visualizes the corresponding anomaly-score profiles, where the horizontal dashed line represents the 99th percentile of the scores in the normal region.

**Cases where both models succeed.**   Figure 7 shows a representative case where both models successfully detect the anomaly. The training and test regions have similar mean levels, and the anomaly appears as a localized point-wise deviation. In the reconstruction panels, both MOMENT and MOMEMTO fail to accurately reconstruct the anomalous spike, which leads to high anomaly scores around the ground-truth anomalous point. At the same time, their scores remain relatively low over the surrounding normal regions. This suggests that sharp local perturbations remain difficult to reconstruct even for the highly generalizable MOMENT backbone.

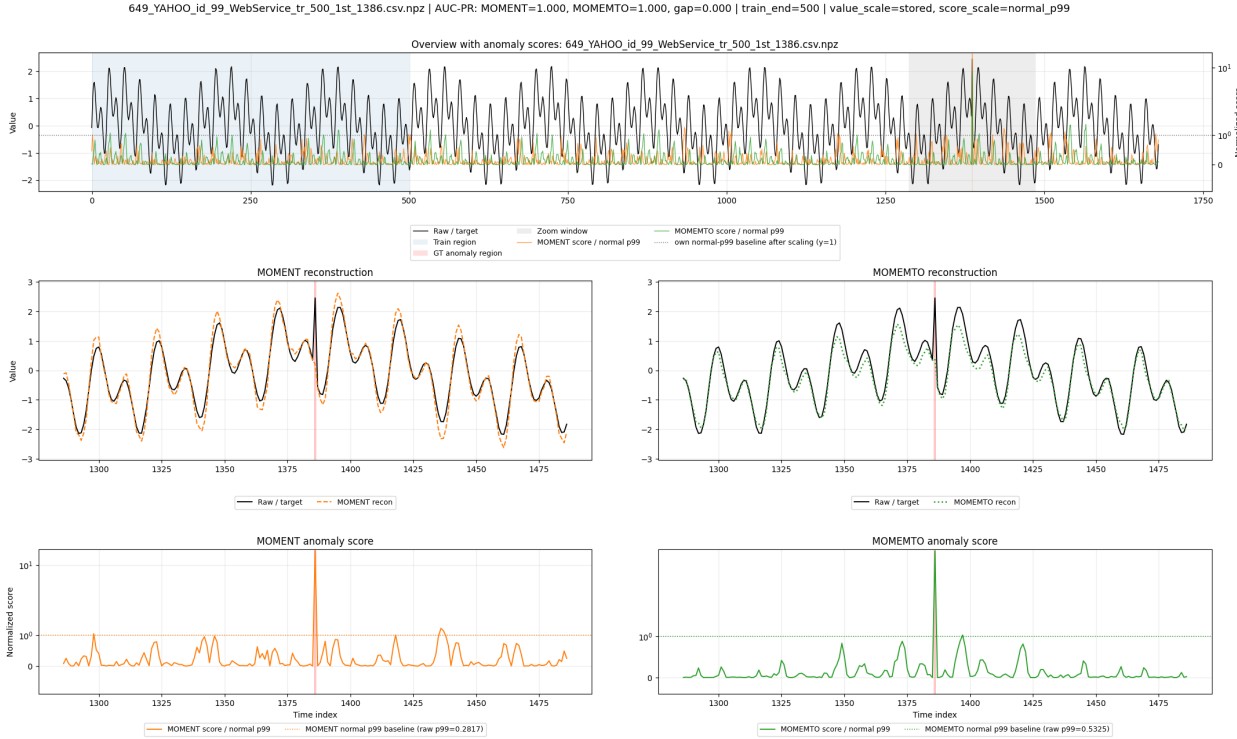

Figure 7: Representative case where both models succeed. MOMENT and MOMEMTO assign high anomaly scores to localized point anomalies while maintaining low scores in normal regions.

**Cases where MOMEMTO succeeds while MOMENT fails.** Figure 8 illustrates cases where the benefit of memory-guided reconstruction becomes more evident. These cases typically involve anomalous intervals with relatively simple temporal structure, including deviations in mean level from the training region. MOMENT often reconstructs these anomalous inputs faithfully, including the shifted mean, and therefore produces small reconstruction errors within the anomalous interval. This suggests that the backbone TSFM can over-generalize to simple out-of-distribution patterns. By contrast, MOMEMTO biases the reconstruction toward normal patterns retrieved from the patch-based memory module. Consequently, the reconstructed output remains closer to the learned normal mean level, increasing the reconstruction discrepancy for the mean-shifted anomalous input and producing clearer separation between normal and anomalous scores.

**Cases where MOMENT succeeds while MOMEMTO fails.** In some cases, MOMEMTO is evaluated as worse than MOMENT, but the qualitative results suggest that these failures are often associated with train–test distribution shift or label ambiguity. Figure 9 shows a representative label-error case. The time-series segment that appears after the red-marked anomalous region is annotated as normal, yet its pattern is clearly inconsistent with the training region, making the normal label questionable. Figure 10 illustrates another case in which the test region is substantially shifted in mean level from the training region. Because MOMEMTO reconstructs inputs through normal patterns stored in memory, it produces higher reconstruction errors over this shifted region. However, when the shifted region is largely labeled as normal, these elevated scores distort the overall anomaly-score distribution. As a result, the actual labeled anomalies are no longer well separated from the high-scoring background, and the detection performance deteriorates. Overall, our qualitative inspection indicates that most of these apparent failure cases are caused by train–test distribution shifts, while a smaller subset is attributable to ambiguous or potentially incorrect labels.

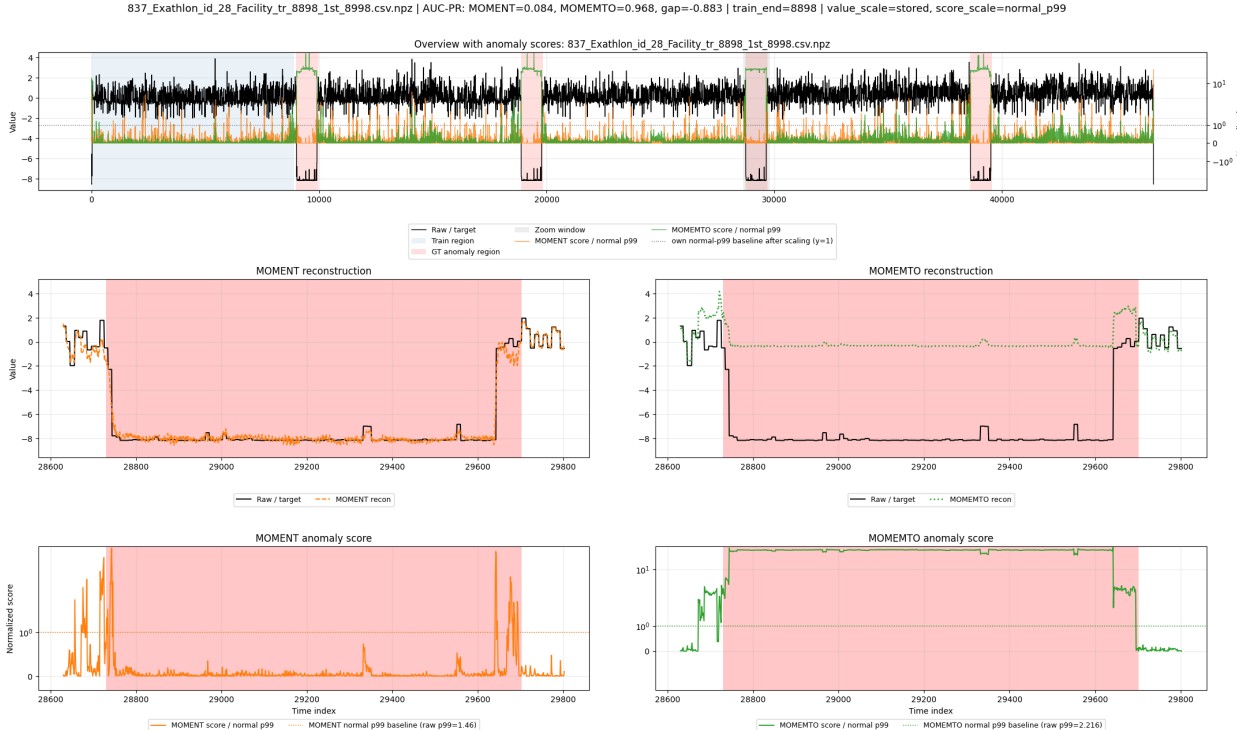

Figure 8: Representative case where MOMEMTO succeeds while MOMENT fails. MOMENT faithfully reconstructs the simple mean-shifted anomaly, whereas MOMEMTO reconstructs it closer to the learned normal pattern, yielding higher anomaly scores.

**Cases where both models fail.** Figure 11 presents a representative example of the dominant failure pattern observed when both MOMENT and MOMEMTO fail to produce clear anomaly-score separation. In this example, the labeled anomalous region does not show a clear deviation from the surrounding normal patterns or from the training distribution, whereas some regions labeled as normal appear more abnormal than the annotated anomalous interval. Similar annotation inconsistencies are observed in most cases in this category, suggesting that the apparent failures are often driven by a mismatch between the benchmark labels and the visual or distributional characteristics of the time series. Consequently, both models assign low or comparable anomaly scores to the labeled anomalous region, resulting in poor detection performance under the given labels.

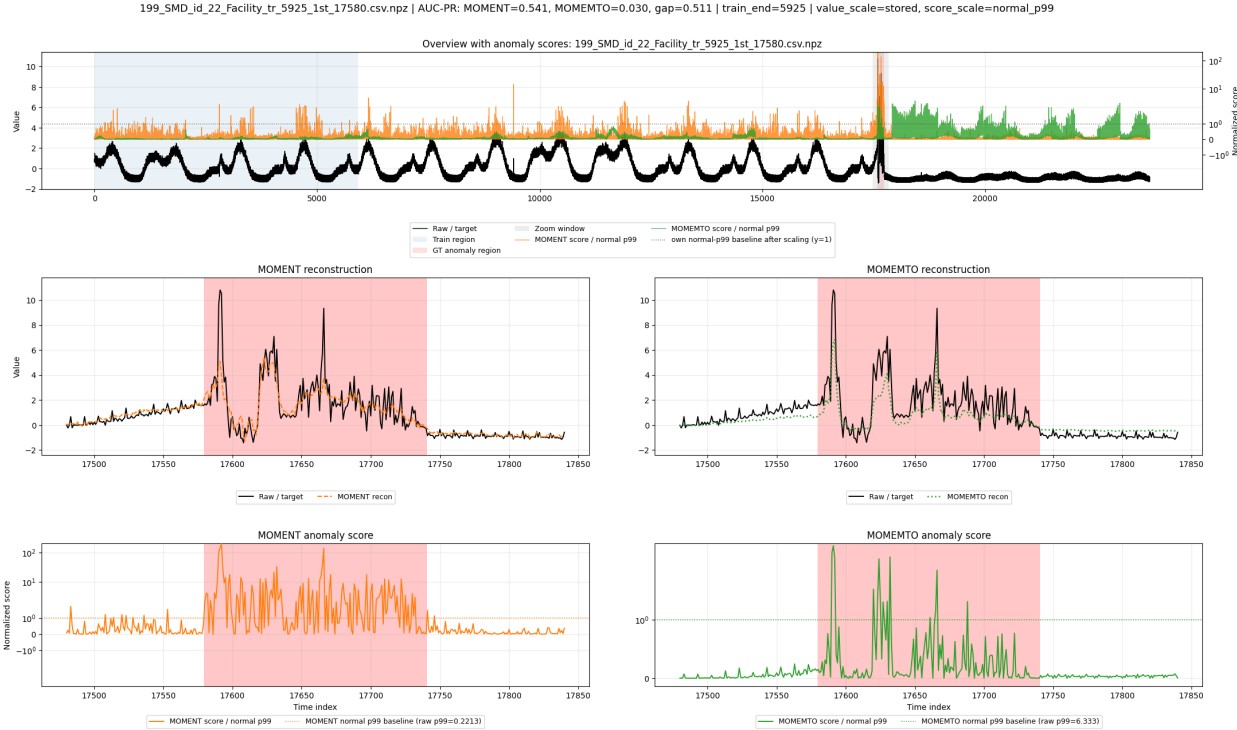

Figure 9: Representative label-error case where MOMENT succeeds while MOMEMTO fails under the benchmark labels. The segment following the red-marked anomaly appears inconsistent with the training region despite being labeled as normal.

## C.2 Number of Referenced Memory Items

We analyze how the number of referenced memory items ($K$) and the training data ratio jointly affect the performance of MOMEMTO. As shown in Table 11, $K = 1$ already yields stable and competitive performance, particularly in the lowest-data regime. Nevertheless, referencing multiple memory items generally improves performance, with the best AUC-PR obtained by $K = 2$–$4$ in most settings. The gains are not monotonic with respect to $K$, indicating that MOMEMTO benefits from a small set of relevant memory items rather than from referencing more items indiscriminately.

Table 11: Few-shot performance sensitivity to the number of referenced memory items (AUC-PR).

| $K$ | 0.1 | 0.2 | 0.3 | 0.4 | 0.6 | 0.8 |
|---|---|---|---|---|---|---|
| 1 | **32.97** | 33.63 | 34.11 | 34.04 | 35.45 | 36.17 |
| 2 | 32.80 | 32.66 | **34.35** | **34.19** | 35.56 | 36.08 |
| 3 | 32.60 | 33.03 | 33.87 | 34.18 | 35.62 | **36.24** |
| 4 | 32.79 | **33.71** | 33.92 | 34.17 | **35.73** | 35.81 |
| 5 | 32.79 | 33.63 | 33.80 | 34.05 | 35.65 | 35.95 |

## C.3 Zero-shot Anomaly Detection

We conduct a leave-one-out evaluation across 32 domains to assess the zero-shot performance of MOMEMTO. In each iteration, one domain is held out as the target domain, while the model is trained on the remaining 31 domains. The trained model is then directly applied to the held-out domain without any fine-tuning, under a zero-shot setting. Performance metrics are computed individually for each target domain and subsequently

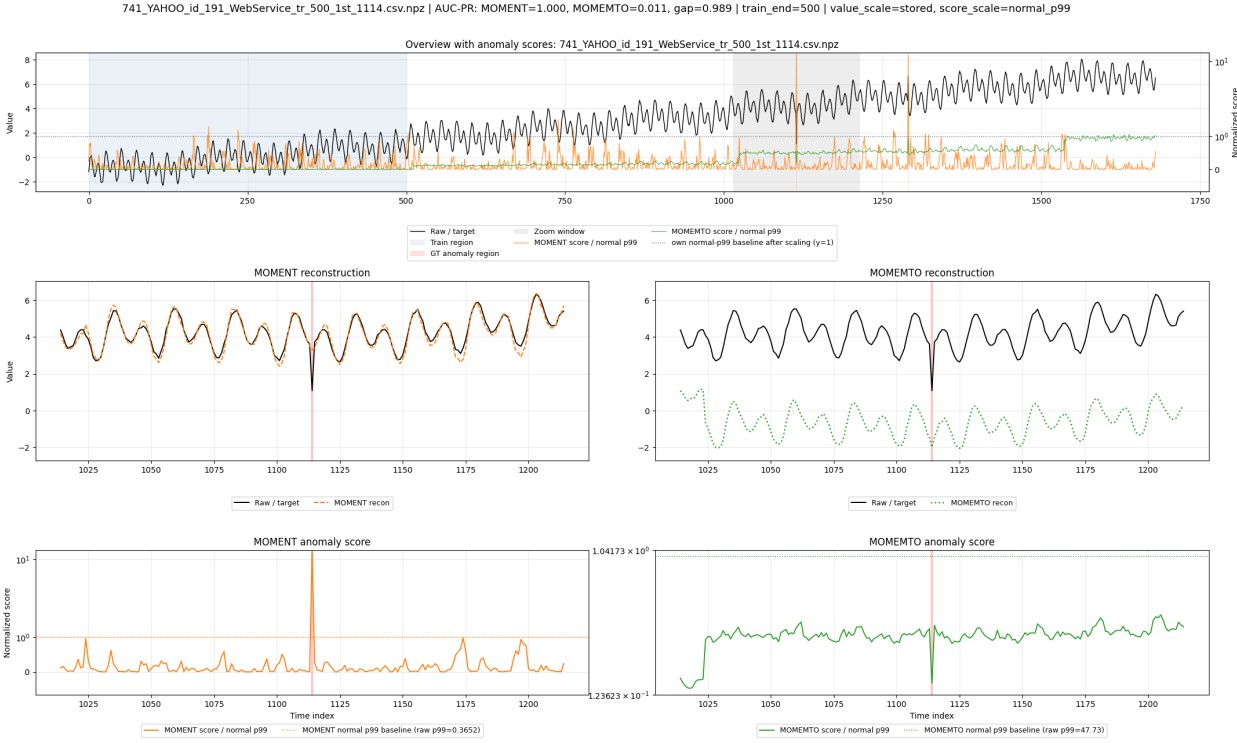

Figure 10: Representative train–test mean-shift case where MOMEMTO is disadvantaged under the benchmark labels. The shifted test region receives elevated anomaly scores, which distorts the score distribution and reduces separation of the labeled anomalies.

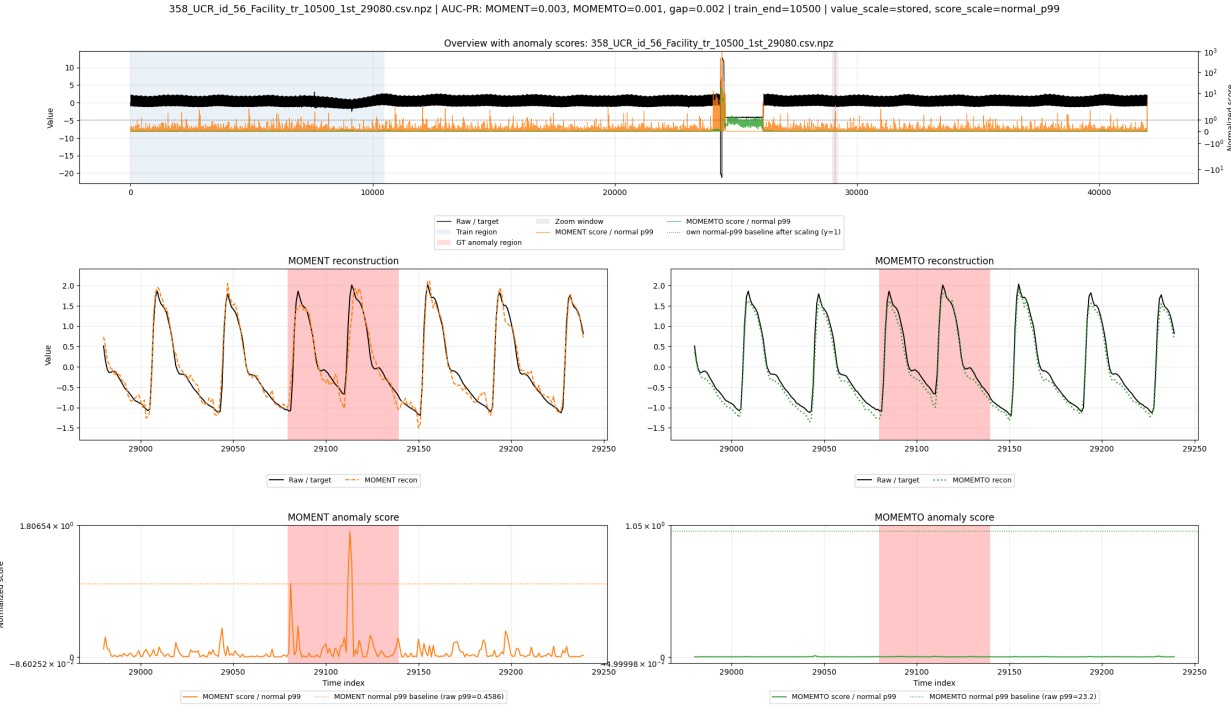

Figure 11: Representative case where both models fail. The labeled anomalous region is visually similar to the surrounding pattern, while some normal-labeled regions appear more abnormal, suggesting annotation inconsistency.

aggregated over all 32 domains. To highlight the effectiveness of our memory-augmented design, we also compare MOMEMTO against its backbone MOMENT.

Table 12: Zero-shot anomaly detection results under leave-one-out evaluation across 32 domains. MOMEMTO is compared with MOMENT.

|  | AUC-PR | AUC-ROC | VUS-PR | VUS-ROC |
|---|---|---|---|---|
| MOMENT$_{md}$ | 29.68 | 69.61 | 30.64 | 76.57 |
| MOMEMTO$_{md}$ | **34.51** | **72.17** | **35.76** | **78.32** |

### C.4 Number of Memory Items

To evaluate how the number of memory items affects MOMEMTO, we conduct two sensitivity analyses. First, we vary the number of memory items from 64 down to 1 on the main benchmark. Second, we evaluate the same factor under few-shot settings by varying the training data ratio. In these experiments, memory items are initialized using K-means centroid values. We also visualize the row-normalized scaled domain-to-memory similarities at the beginning and end of training for different memory sizes.

As shown in Table 13, MOMEMTO maintains stable performance across a wide range of memory sizes on the main benchmark. Although the best AUC-PR is obtained with (M=16), the performance changes only moderately as the number of memory items decreases, suggesting that the model does not strongly depend on a large memory capacity. Table 14 shows a similar trend under few-shot settings, where small memory sizes remain competitive across different training-data ratios.

Figure 12 further shows that memory access tends to become concentrated on a subset of memory items after training. This tendency is consistently observed across different memory sizes, suggesting that the data-driven update mechanism encourages frequently accessed memory items to serve as reusable normal-pattern prototypes.

Table 13: Sensitivity analysis of the number of memory items on the main benchmark.

| M | 64 | 32 | 16 | 8 | 4 | 2 | 1 |
|---|---|---|---|---|---|---|---|
| AUC-PR | 36.11 | 36.05 | 36.26 | 36.07 | 35.97 | 35.69 | 35.81 |
| AUC-ROC | 74.59 | 74.22 | 74.18 | 74.19 | 74.30 | 73.25 | 73.50 |

Table 14: Few-shot performance sensitivity to the number of memory items (AUC-PR).

| # Memory items | 0.1 | 0.2 | 0.3 | 0.4 | 0.6 | 0.8 |
|---|---|---|---|---|---|---|
| 16 | 33.24 | **33.06** | 33.91 | 34.00 | 35.29 | 36.25 |
| 8 | 33.51 | 33.03 | 34.31 | 33.98 | **35.31** | **36.30** |
| 4 | 33.51 | 32.88 | 34.38 | 34.05 | 35.10 | 35.88 |
| 2 | 33.57 | 32.76 | 34.50 | 33.94 | 35.04 | 35.93 |
| 1 | **33.89** | 32.43 | **34.52** | **34.14** | 34.89 | 36.17 |

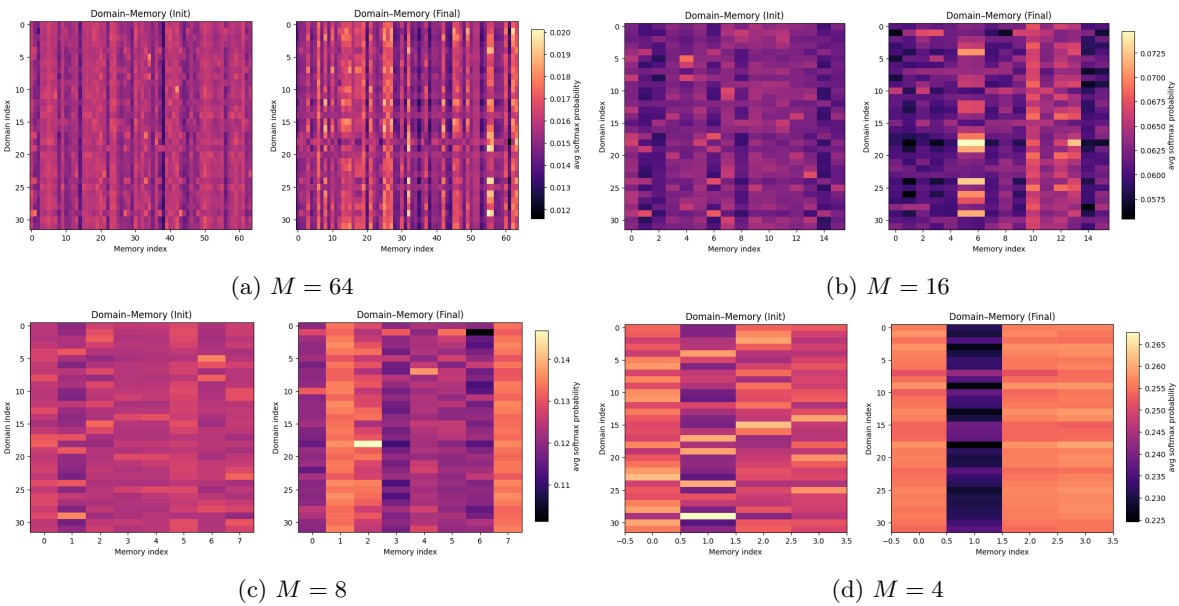

Figure 12: Scaled domain-to-memory similarity maps at the beginning (left) and end of training (right) for different numbers of memory items. Each heatmap is row-normalized.

## C.5 Decoder Input Ablation

We conduct an additional ablation to examine the effect of the decoder input representation. In the default MOMEMTO architecture, the decoder receives the concatenated representation $[\mathbf{q}; \tilde{\mathbf{q}}]$, where $\mathbf{q}$ denotes the original encoder query and $\tilde{\mathbf{q}}$ denotes the memory-refined query. We compare this default setting with two variants that use only $\mathbf{q}$ or only $\tilde{\mathbf{q}}$ as the decoder input.

Table 15: Decoder input ablation study.

| Decoder input | AUC-PR | AUC-ROC |
|---|---|---|
| Original query only ($\mathbf{q}$) | 29.58 | 69.33 |
| Memory-refined query only ($\tilde{\mathbf{q}}$) | 32.58 | 69.54 |
| Concatenation ($[\mathbf{q}; \tilde{\mathbf{q}}]$) | 36.35 | 74.83 |

## C.6 Additional Comparison with Similar Input Window Lengths

To provide an additional comparison under similar input lengths, we report the performance of TSFM-based models using window lengths around 96–100. As shown in Table 16, MOMEMTO still achieves the highest AUC-PR and AUC-ROC among the compared TSFM-based models. This result indicates that the performance gain of MOMEMTO is not solely attributable to using a longer input window in the main experiments.

Table 16: Additional comparison of TSFM-based models under similar input window lengths.

| Model | Win. len. | AUC-PR | AUC-ROC |
|---|---|---|---|
| Chronos$_{md}$ | 100 | 26.43 | 65.48 |
| TimesFM$_{md}$ | 96 | 26.67 | 66.95 |
| MOMENT$_{md}$ | 96 | 26.71 | 67.27 |
| DADA$_{md}$ | 100 | 33.08 | 70.04 |
| **MOMEMTO$_{md}$** | **96** | **34.90** | **73.00** |

## C.7 Patch Size Sensitivity

Since the pre-trained MOMENT backbone follows the default patch configuration with patch length 8, we evaluate patch-size sensitivity under **the scratch encoder** setting to avoid changing the patch configuration of the pre-trained backbone.

Table 17: Sensitivity analysis of patch length under the scratch encoder setting.

| Patch length $L$ | AUC-PR | AUC-ROC |
|---|---|---|
| 1 | 28.87 | 67.52 |
| 2 | 30.71 | 70.24 |
| 4 | 32.69 | 70.68 |
| 8 | 33.63 | 72.17 |
| 16 | **34.08** | **72.64** |

## C.8 Training and Inference Time

We measure the training time, inference time, and number of parameters under different configurations of the patch-based memory module (PMM) and multi-domain training, as summarized in Table 18. The multi-domain configurations train a single model, resulting in 0.388B parameters for MOMEMTO and 0.385B parameters for the variant without PMM. For configurations without multi-domain training, separate models are trained for each domain, and the reported parameter counts correspond to the aggregate over all trained domain models. Under this setting, the aggregate number of parameters is 12.42B for MOMEMTO and 12.32B for the variant without PMM. The inference time is similar across configurations, ranging from 55.15 to 59.22 seconds. The reported training time is 92.11 seconds for MOMEMTO and 59.42 seconds for the variant without PMM in the multi-domain setting. Without multi-domain training, the training time is 224.12 seconds for MOMEMTO and 185.06 seconds for the variant without PMM.

## C.9 Runtime scaling with memory and sequence length

To assess the practical cost of the patch-based memory module, we measure per-batch training time, inference time, and peak GPU memory while varying the number of memory items ($M$), the number of retrieved memory items ($K$), and the number of input patches ($N$). All measurements are conducted under the same full fine-tuning setting with a fixed batch size and three repeated runs. Training time includes the forward pass, loss computation, backward pass, and optimizer update, while inference time is measured without gradient computation.

As shown in Table 19, increasing $M$ from 4 to 128 has limited effect on per-batch runtime, with training time remaining within 137–143 ms and inference time within 21.78–21.99 ms. Similarly, increasing $K$ from

Table 18: Training time, inference time, and number of parameters under different configurations of the patch-based memory module (PMM) and multi-domain training. The configuration without PMM is equivalent to MOMENT.

| Configuration | Time (s) | | # Params (B) |
|---|---|---|---|
| | Training | Inference | |
| MOMEMTO | 92.11 | 59.22 | 0.388 |
| w/o PMM | 59.42 | 55.15 | 0.385 |
| w/o multi-domain training | 224.12 | 58.91 | 12.42 |
| w/o PMM & multi-domain training | 185.06 | 55.49 | 12.32 |

1 to 16 results in only a small runtime change. The clearest scaling trend is observed with respect to the number of patches $N$: increasing $N$ from 8 to 64 increases training time from 107.21 ms to 137.19 ms per batch and peak training memory from 7.67 GB to 9.96 GB. These results indicate that, within the tested range, the runtime overhead is more sensitive to the input length than to the number of memory items or retrieved items.

Table 19: Runtime and GPU memory scaling of MOMEMTO with respect to the number of memory items ($M$), retrieved items ($K$), and input patches ($N$). Training time is measured per batch and includes the forward pass, loss computation, backward pass, and optimizer update. Inference time is measured per batch without gradient computation. We use patch length $L = 8$ and embedding dimension $d_{\text{model}} = 1024$; thus, the window length is $N \times L$. Values are averaged over three runs, with standard deviations reported for runtime.

| Configuration | | | Measurements | | | |
|---|---|---|---|---|---|---|
| $M$ | $K$ | $N$ | Train time (ms) | Infer time (ms) | Train mem. (GB) | Infer mem. (GB) |
| 4 | | | $142.81 \pm 5.20$ | $21.82 \pm 0.03$ | 9.81 | 2.38 |
| 8 | | | $137.77 \pm 1.00$ | $21.84 \pm 0.11$ | 9.83 | 2.83 |
| 16 | 3 | 64 | $138.63 \pm 2.89$ | $21.78 \pm 0.02$ | 9.88 | 2.84 |
| 32 | | | $139.49 \pm 3.93$ | $21.83 \pm 0.02$ | 9.96 | 2.85 |
| 64 | | | $139.44 \pm 3.93$ | $21.87 \pm 0.08$ | 10.12 | 2.87 |
| 128 | | | $137.48 \pm 1.00$ | $21.99 \pm 0.03$ | 10.41 | 2.92 |
| | 1 | | $135.15 \pm 0.80$ | $21.79 \pm 0.02$ | 9.91 | 2.85 |
| | 2 | | $138.28 \pm 3.68$ | $21.85 \pm 0.13$ | 9.93 | 2.85 |
| 32 | 3 | 64 | $137.26 \pm 1.59$ | $21.97 \pm 0.12$ | 9.96 | 2.85 |
| | 4 | | $140.54 \pm 1.70$ | $21.91 \pm 0.11$ | 9.98 | 2.85 |
| | 8 | | $141.81 \pm 2.28$ | $21.95 \pm 0.01$ | 10.09 | 2.90 |
| | 16 | | $137.01 \pm 0.47$ | $22.18 \pm 0.03$ | 10.28 | 3.00 |
| | | 8 | $107.21 \pm 0.38$ | $17.32 \pm 0.01$ | 7.67 | 2.77 |
| 32 | 3 | 16 | $108.52 \pm 0.44$ | $17.22 \pm 0.03$ | 7.67 | 2.78 |
| | | 32 | $115.25 \pm 0.21$ | $17.51 \pm 0.03$ | 8.15 | 2.81 |
| | | 64 | $137.19 \pm 0.28$ | $21.84 \pm 0.03$ | 9.96 | 2.85 |

### C.10 Quantitative Analysis of Over-generalization

In reconstruction-based anomaly detection, over-generalization causes anomalous inputs to be reconstructed too well, making their anomaly scores overlap with those of normal samples. Therefore, clearer separation between the score distributions of normal and anomalous samples is consistent with reduced over-generalization.

To quantify this effect, we computed the two-sample Kolmogorov–Smirnov (KS) statistic for each time series. The KS statistic measures the maximum difference between the cumulative distributions of normal and anomalous scores, which reflects the degree of separation between the two score distributions. A larger KS value indicates a clearer separation between normal and anomalous scores.

Table 20: Comparison of the Kolmogorov–Smirnov (KS) statistic between normal and anomalous anomaly-score distributions for original TSFMs and their PMM-augmented variants under multi-domain training. Higher KS values indicate clearer separation between normal and anomalous samples, suggesting reduced over-generalization.

| Model | Original TSFM | TSFM + PMM |
|---|---|---|
| | KS | KS |
| $\text{UP2ME}_{md}$ | 0.4178 | **0.4687** |
| $\text{UniTS}_{md}$ | 0.3547 | **0.5020** |
| $\text{MOMENT}_{md}$ | 0.4270 | **0.5343** |

### C.11 Qualitative Analysis of Over-generalization

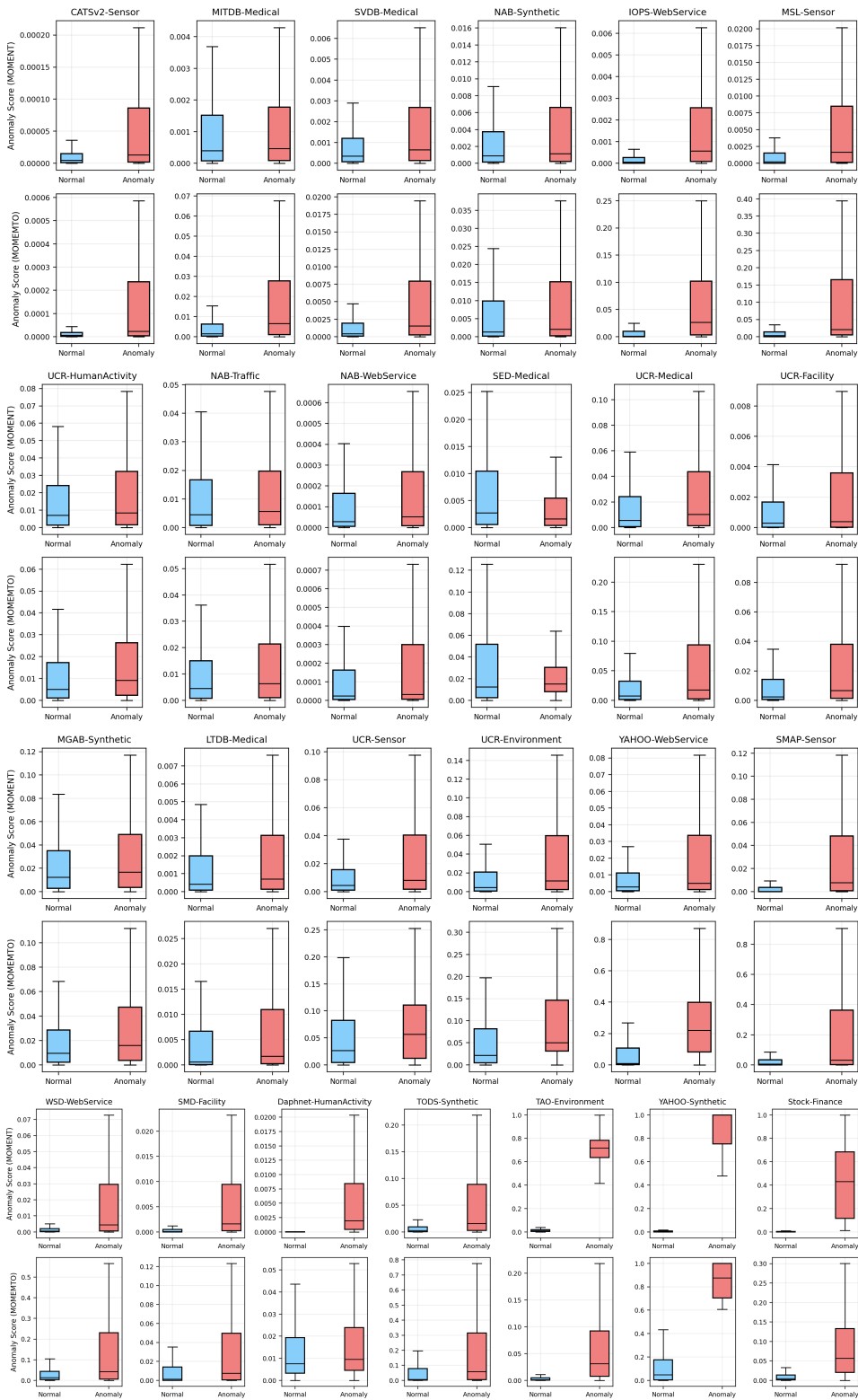

Figure 13: Comparison of anomaly score distributions between normal (blue) and anomalous (red) samples across a subset of domains. The upper row shows the results obtained with the backbone model MOMENT, while the lower row corresponds to our proposed model.

