# OpenReview forum: "Patch-based Memory Gate Model in Time Series Foundation Model"
_TMLR — Decision pending for TMLR_

### Review · Reviewer_cNkZ · 2026-05-02

**Summary Of Contributions:**

The paper proposes MOMEMTO, a reconstruction-based anomaly detector that combines the pre-trained MOMENT encoder with a patch-based memory module inspired by MEMTO. The memory stores patch-level normal-pattern representations, retrieves the top-K similar memory items, updates them with gated attention, and uses the retrieved representation to bias reconstruction toward normal patterns. The paper also proposes multi-domain joint fine-tuning, where a single model is trained across the TSB-AD-U univariate benchmark rather than maintaining a separate model for each series. Empirically, MOMEMTO achieves the best aggregate AUC-PR, AUC-ROC, VUS-PR, and VUS-ROC among the reported baselines, improves over MOMENT and DADA on the main benchmark, and includes useful ablations for the pre-trained encoder, patch-based memory, memory update strategy, memory count, few-shot learning, zero-shot leave-one-domain-out performance, and compatibility with UniTS/UP2ME. The main weaknesses are that several central claims are stronger than the evidence currently supports: comparison fairness and aggregation are not sufficiently clear, there is no uncertainty analysis despite modest aggregate gains over strong baselines, the memory-update mechanism is underspecified from a reproducibility perspective, and the scope is limited to a single univariate benchmark while the paper makes broader claims about time series foundation models and multi-domain anomaly detection.

**Additional Comments:**

The paper has a promising core idea and a useful empirical package, but I would not consider the current version ready without more rigorous evidence and clearer reproducibility details. The most important fixes are not cosmetic: the authors need to make the evaluation protocol auditable, quantify robustness, and clarify the exact memory update mechanism.

**Audience:**

Yes

**Audience Explanation:**

The problem is relevant to the TMLR audience. Adapting time series foundation models to anomaly detection is timely, and the tension between strong general-purpose representations and reconstruction over-generalization is a real issue. A memory-augmented TSFM that can be jointly fine-tuned across heterogeneous domains would be practically useful if the evidence holds up. The paper will be most interesting to researchers working on time series foundation models, unsupervised anomaly detection, memory/prototype mechanisms, and cross-domain adaptation. The current contribution is more of a well-motivated architectural adaptation and empirical study than a major conceptual advance, but the experimental direction is worthwhile.

**Broader Impact Concerns:**

The paper concerns anomaly detection in cyber-physical and operational monitoring systems. False negatives could miss safety-critical or security-relevant failures, while false positives could cause costly operational interventions. The paper should briefly discuss these risks, especially because the evaluation is threshold-independent and does not address how the method behaves under deployment thresholds, alert fatigue, or domain shift. I do not see unusual dual-use concerns beyond the standard risks of automated monitoring and anomaly detection.

**Claims And Evidence:**

No

**Claims Explanation:**

The empirical results are directionally encouraging. The main table shows MOMEMTOmd outperforming all reported baselines on aggregate metrics, including DADAmd and MOMENTmd, and the ablation table supports the claim that both the pre-trained encoder and the patch-based memory module contribute to performance. The few-shot, zero-shot, and TSFM-compatibility experiments are useful additions, and the appendix provides more detail than many papers in this area.

However, the evidence is not yet convincing enough for the paper's strongest claims. First, no standard deviations, confidence intervals, statistical tests, or seed sensitivity are reported. This matters because the main aggregate gains over the strongest baselines are meaningful but not overwhelming, and the per-domain VUS-PR table shows large domain-level variability with several domains where MOMEMTOmd is worse than MOMENT, DADA-like alternatives, or simpler baselines. The paper needs to report uncertainty across runs and ideally paired comparisons across time series or domains.

Second, the benchmark aggregation protocol is not clear enough. It is unclear whether metrics are macro-averaged across series, macro-averaged across domains, weighted by sequence length, or aggregated another way. Since TSB-AD-U contains highly heterogeneous datasets with very different anomaly ratios, sequence lengths, and numbers of series, this choice can materially change conclusions. The paper should also explain exactly how train/test splits, normalization, and subsequence construction are handled for every model, and whether the multi-domain model ever receives information that is unavailable to the corresponding single-domain baselines.

Third, the comparison between multi-domain MOMEMTO and the "one-model-per-dataset" setting is confusing. Appendix C.2 says the Eval set has 350 time series and the conventional scheme trains 350 independently trained models, which is closer to one-model-per-series than one-model-per-dataset. This inflates the stated model-size and context-switching contrast and should be corrected or justified. The compute comparison in Table 4 should include a fair deployment scenario and should avoid presenting the sum of hundreds of checkpoints as the only relevant alternative.

Fourth, the memory mechanism is not fully reproducible from the description. The paper alternates between treating memory items as stored state, initialized prototypes, and updated values inside the forward pass. It should specify whether memory items are learnable parameters, persistent buffers updated online, per-batch temporary states, or outputs of differentiable transformations; how updates are accumulated across batches and epochs; whether update order affects results; and what happens during inference. This is central to the method.

Fifth, some key baselines and comparisons are missing or under-justified. H-PAD is discussed as a closely related patch/prototype method but is not included in the experiments or clearly ruled out as inapplicable. Since the paper claims an advantage from patch-level memory for interval or periodic anomalies, this is a particularly relevant comparison. The paper should also clarify whether the reported implementations of DADA, MOMENT, UniTS, UP2ME, Chronos, and TimesFM use comparable fine-tuning budgets and input windows.

Finally, the claim that the method mitigates over-generalization is plausible but only indirectly supported. Better anomaly-score separation and improved AUC are consistent with reduced over-generalization, but the paper should more directly show reconstruction behavior on anomalous inputs, memory retrieval examples, or counterfactual comparisons demonstrating that the memory module prevents faithful reconstruction of abnormal patterns.

**Requested Changes:**

(Critical) Add uncertainty analysis and statistical support for the main empirical claims. Report means and standard deviations over multiple seeds, confidence intervals or paired significance tests across time series/domains, and discuss cases where MOMEMTOmd underperforms strong baselines in the per-domain results.

(Critical) Clarify the evaluation and aggregation protocol. State exactly how AUC/VUS metrics are averaged, how train/test splits and normalization are performed, how subsequences are generated for each model, and how multi-domain training avoids test leakage or unfair access to target-domain information.

(Critical) Correct or justify the "one-model-per-dataset" terminology and the compute/model-size comparison. If 350 models are trained for 350 series, call this one-model-per-series and explain why that is the right baseline. Add a comparison against one model per dataset/source or another practical deployment setup.

(Critical) Specify the memory-state semantics precisely. Explain whether memory items are parameters, buffers, or temporary activations; how updates persist across batches; whether training is order-dependent; how gradients flow through memory updates; and what memory is used at inference time.

(Critical) Include or justify omission of closely related prototype/memory baselines, especially H-PAD, and clarify whether other TSFM baselines receive comparable fine-tuning, context length, training budget, and data access.

(Critical) Scope the claims to the demonstrated setting or add broader evidence. The current experiments are on one univariate benchmark, so claims about general time series anomaly detection, multi-domain TSFMs, and broad applicability should either be softened or supported with multivariate and additional real-world benchmarks.

(Minor) Directly evaluate over-generalization. Add reconstruction examples, normal/anomaly reconstruction-error breakdowns, memory retrieval visualizations for anomalous inputs, or an experiment where the decoder's ability to reconstruct anomalies is explicitly measured.

(Minor) Improve the method presentation. The equations for memory update and query update are hard to follow because dimensions, softmax axes, and the roles of q/m/v/w are not always explicit. A compact tensor-shape table would make the method easier to reproduce.

(Minor) Report runtime and memory usage in a way that separates pre-training, fine-tuning, inference, checkpoint loading, and parameter count per deployed model. The current context-switching and 455 GB checkpoint numbers are not sufficiently interpretable.

(Minor) Strengthen writing consistency. Use one term consistently for "dataset," "domain," "time series," and "sub-domain," and avoid broad statements such as "superior performance" unless the statement is tied to the exact benchmark and metric where it is demonstrated.

---

> ### Author Response · Authors · 2026-06-16
>
> We are grateful to Reviewer cNkZ for the thorough and rigorous assessment of our work. The detailed comments helped us substantially improve the statistical support, evaluation protocol, terminology, and overall framing of the empirical claims in the revised manuscript.
>
> >(Critical) Add uncertainty analysis and statistical support for the main empirical claims. Report means and standard deviations over multiple seeds, confidence intervals or paired significance tests across time series/domains, and discuss cases where MOMEMTOmd underperforms strong baselines in the per-domain results.
>
>
> To address this point, we have updated **Table 2 and Figure 2 in Section 4.1 (Main Results)** to report the mean and standard deviation over five random seeds.
>
>  In addition, we have added **Appendix C.1 (Qualitative Case Analysis of MOMENT and MOMEMTO)**, which compares representative success and failure cases of MOMEMTO and its backbone TSFM, MOMENT. This analysis discusses both cases where the memory module improves anomaly detection and cases where MOMEMTO exhibits limitations, such as train–test distribution shifts or ambiguous anomaly labels.
>
> >(Critical) (Critical) Clarify the evaluation and aggregation protocol. State exactly how AUC/VUS metrics are averaged, how train/test splits and normalization are performed, how subsequences are generated for each model, and how multi-domain training avoids test leakage or unfair access to target-domain information.
>
>
> We have revised the **Experimental setup and implementation details paragraphs in Section 4 (Experiments)** and **Appendix B.3 (Baselines and Hyperparameter Settings)** to make the evaluation protocol more explicit. The revised description summarizes the benchmark split, normalization fitted only on the training segments, and model-specific subsequence/window generation procedures. We have also clarified that each metric is computed independently for each of the 350 Eval time series and then averaged across the series.
>
> >(Critical) Correct or justify the "one-model-per-dataset" terminology and the compute/model-size comparison. If 350 models are trained for 350 series, call this one-model-per-series and explain why that is the right baseline. Add a comparison against one model per dataset/source or another practical deployment setup
>
>
> We agree with the reviewer’s concern and have revised the experimental setting accordingly. In the previous version, the non-multi-domain comparison used a one-model-per-series setup, where a separate model was trained for each individual evaluation time series. In the revised manuscript, we have replaced this with a one-model-per-domain setup: the 350 Eval time series are grouped into 32 domains based on the source dataset and sub-domain metadata, and a separate model is trained for each domain.
>
> We have rerun the main experiments for all relevant baselines under this revised setting. This provides a more practical and stronger comparison than the previous one-model-per-series setup. We have also updated the terminology (ex. one-model-per-dataset -> one-model-per-domain), experimental setup paragraph in Section 4 (Experiments), and compute/model-size comparison to reflect the 32-domain setting in **Appendix C.8 (Training and Inference Time)**
>
> >(Critical) Specify the memory-state semantics precisely. Explain whether memory items are parameters, buffers, or temporary activations; how updates persist across batches; whether training is order-dependent; how gradients flow through memory updates; and what memory is used at inference time
>
>
> We have clarified the memory-state semantics by adding a new **Memory dynamics paragraph at the end of Section 3.1.2 (Patch-based Memory Module) in the Methodology section**. The revised paragraph explicitly states that memory items are persistent state variables, not parameters directly optimized by gradients. During training, they are updated online at the sample level within each mini-batch using the gated update rule, and the updated memory state is carried over to subsequent samples and batches, making memory evolution order-dependent.
>
> We have also clarified that gradients from the reconstruction loss update the encoder, decoder, and the projection matrices controlling the memory update gate, whereas the memory values themselves are revised by the explicit update rule. At inference time, the final trained memory is kept fixed and used only for memory retrieval and query refinement, without test-time memory updates.

---

> ### Author Response · Authors · 2026-06-16
>
> >(Critical) Include or justify omission of closely related prototype/memory baselines, especially H-PAD, and clarify whether other TSFM baselines receive comparable fine-tuning, context length, training budget, and data access.
>
>
> H-PAD is a closely related memory-based reconstruction method for time series anomaly detection. However, we have not included it in the quantitative comparison because, to the best of our knowledge at the time of revision, its official implementation is not publicly available and the paper does not provide sufficient details for several key hyperparameters needed for faithful reproduction.
>
> To improve the reproducibility and fairness of the evaluated baselines, we have revised **Appendix B.3 (Baselines and Hyperparameter Settings)** to provide clearer descriptions of the baseline hyperparameters and model-specific subsequence/window generation procedures. We have also added **Appendix C.6 (Additional Comparison with Similar Input Window Lengths)**, including Table 16, to compare TSFM baselines under comparable window lengths and support a fairer evaluation.
>
> (Table 16 in Appendix C.6)
> | Model | Win. len. | AUC-PR | AUC-ROC |
> |---|---:|---:|---:|
> | Chronos$_{md}$ | 100 | 26.43 | 65.48 |
> | TimesFM$_{md}$ | 96 | 26.67 | 66.95 |
> | MOMENT$_{md}$ | 96 | 26.71 | 67.27 |
> | DADA$_{md}$ | 100 | 33.08 | 70.04 |
> | **MOMEMTO$_{md}$** | **96** | **34.90** | **73.00** |
>
> >(Critical) Scope the claims to the demonstrated setting or add broader evidence. The current experiments are on one univariate benchmark, so claims about general time series anomaly detection, multi-domain TSFMs, and broad applicability should either be softened or supported with multivariate and additional real-world benchmarks.
>
> We have revised the manuscript to avoid overstating the applicability of the proposed method beyond the demonstrated setting.
>  In particular, we have clarified that the current study focuses on univariate time series anomaly detection and have softened broader claims about general time series anomaly detection and broad applicability.
>
> Specifically, we revised **the contribution summary at the end of the Introduction** to explicitly state the univariate setting and the use of 23 univariate benchmark datasets. We have also updated **Section 5 (Conclusion)** to acknowledge the univariate scope as a limitation and to discuss extensions to multivariate settings as an important direction for future work.
>
> >(Minor) Directly evaluate over-generalization. Add reconstruction examples, normal/anomaly reconstruction-error breakdowns, memory retrieval visualizations for anomalous inputs, or an experiment where the decoder's ability to reconstruct anomalies is explicitly measured.
>
> We have added a controlled reconstruction-behavior analysis in **Section 4.2 (Model Analysis), under the paragraph titled “Over-generalization alleviation,”** to more directly examine over-generalization. This analysis, including Figures 4 and 5, compares the reconstruction behavior of MOMENT, the TSFM backbone of our method, and MOMEMTO on anomalous inputs with simple temporal structures. The results show that MOMENT can faithfully reconstruct such anomalous patterns, whereas MOMEMTO tends to reconstruct them closer to the learned normal range. This observation is consistent with and further supports the success/failure case analysis added in **Appendix C.1 (Qualitative Case Analysis of MOMENT and MOMEMTO).**
>
> >(Minor) Improve the method presentation. The equations for memory update and query update are hard to follow because dimensions, softmax axes, and the roles of q/m/v/w are not always explicit. A compact tensor-shape table would make the method easier to reproduce
>
> We have revised **Section 3.1.2 (Patch-based Memory Module) in the Methodology section** to make the memory update and query update procedures easier to follow. Specifically, we have added Table 1, a compact notation summary that reports the dimensions and descriptions of the main variables used in the memory mechanism, including the query, memory items, selected memory items, update representations, retrieval weights, and memory-refined query.
>
> We have also decomposed the previously condensed memory mechanism into clearer procedural steps, including memory selection, memory update, query update, query alignment, and memory dynamics. In particular, the newly added **memory selection paragraph** explicitly describes the sequence-level similarity computation, Top-K retrieval from raw similarity scores, and softmax normalization over the selected Top-K memory items. We further updated Appendix B.4 (Algorithms), especially Algorithm 2, to align the algorithmic description with these revised explanations.

---

> ### Author Response · Authors · 2026-06-16
>
> >(Minor) Report runtime and memory usage in a way that separates pre-training, fine-tuning, inference, checkpoint loading, and parameter count per deployed model. The current context-switching and 455 GB checkpoint numbers are not sufficiently interpretable.
>
> The original experiment was intended to highlight the efficiency of the proposed multi-domain setting. However, after revising the comparison from a one-model-per-series setup to a one-model-per-domain setup, we considered it more appropriate to soften this claim.
>
> Accordingly, we have removed metrics that are highly dependent on the execution environment or difficult to interpret, such as context-switching time and aggregate checkpoint size. Instead, we now report the overall fine-tuning time, inference time, and parameter count under the revised 32-domain setting in **Appendix C.8 (Training and Inference Time)**, including Table 18. We have also moved this analysis from the main text to the appendix so that it serves as supporting evidence rather than a central efficiency claim.
>
> In addition, we have added a runtime scaling analysis in **Appendix C.9 (Runtime Scaling with Memory and Sequence Length)**, including Table 19. In this analysis, we measure per-batch training time, inference time, and peak GPU memory while varying the number of memory items (M), the number of retrieved memory items (K), and the number of input patches (N). This provides a more controlled view of the computational cost of the proposed memory module under different architectural settings.
>
> >(Minor) Strengthen writing consistency. Use one term consistently for "dataset," "domain," "time series," and "sub-domain," and avoid broad statements such as "superior performance" unless the statement is tied to the exact benchmark and metric where it is demonstrated.
>
> We have revised the manuscript to improve terminology consistency throughout the paper. Following the revised experimental setting, we consistently use “domain” when referring to the training/evaluation grouping used for one-model-per-domain experiments, and distinguish it from “dataset” and individual “time series” where needed. In particular, we clarified the domain definition and training settings in **Section 4 (Experimental setup) and Appendix B.1–B.2**, and updated the caption and description of Table 2 accordingly.
>
> We have also softened broad performance claims, such as “superior performance,” and revised them to be tied to the specific benchmark, metrics, and experimental settings where the improvements are demonstrated, including the **Abstract, the contribution summary at the end of the Introduction, and Section 4.1 (Main Results).**

---

### Review · Reviewer_MbtF · 2026-05-03

**Summary Of Contributions:**

The paper addresses the issue of over-generalization in reconstruction-based anomaly detection models, where the models accurately reconstruct the anomalies at test time, thus unable to predict them. To fix this, the authors propose MOMEMTO, which has a novel memory module designed to store patch level features of normal time series data (non-anomaly). This module utilizes a query-based attention mechanism to retrieve and selectively update memory items, pushing reconstructed ouptuts towards normal patterns. The paper moves away from training a separate model for each dataset. Instead, it fine-tunes one shared model on multiple datasets together. The memory items store common normal patterns from different domains, which helps reduce training cost and use the data more efficiently
The authors show extensive evaluation on the TSB-AD-U benchmark, covering 870 univariate time series from 23 datasets, showing that MOMEMTO achieves the best AUC-PR, AUC-ROC, etc among the baselines considered in the paper.

Strengths:

1. By using multi-domain training, MOMEMTO avoids training and storing many separate models, which the paper shows would create large context-switching overhead and a much larger checkpoint footprint. This reduces total total training time and model footprint.
2. The paper provides useful ablations that separate the contributions of the pre-trained encoder and the patch-based memory module. It also includes anomaly-score distribution visualizations showing that MOMEMTO often creates clearer separation between normal and anomalous samples, which supports the authors’ claim that the memory module helps reduce over-generalization.

Weaknesses:

1. The paper only considers univariate datasets for evals, it will be good to show some initial results on multivariate setting as well.
2. The paper lacks scaling experiments showing how training / inference time, and memory usage change as the memory size and number of retrieved memory items increase. Table 4 provides useful point estimates for one configuration, but it does not show how the method scales with important hyperparams like memory size and number of retrieved items. This makes it harder to assess the practical cost of applying the method to larger memories or longer input windows.
3. The paper lacks clean ablations on comparing point-level and patch level with the same pre-trained backbone. In Table 14 they compare MEMTO and MOMEMTO but the latter has a different pretrained backbone so it is hard to verify whether patch-level memory is more benefecial than point-level

**Audience:**

Yes

**Audience Explanation:**

The paper will be useful for audience in the field of time series analysis and anomaly detection.

**Claims And Evidence:**

Yes

**Claims Explanation:**

The authors provide detailed experiments and ablations to support their claims. The authors show extensive evaluation on the TSB-AD-U benchmark, covering 870 univariate time series from 23 datasets, showing that MOMEMTO achieves the best AUC-PR, AUC-ROC, etc among the baselines considered in the paper. The authors also show that their multi-domain training strategy drastically reduces computational overhead in Table 4, which shows the massive reduction in context-switching time and total parameter footprint (from over 450 GB down to 1.3 GB) when moving from a one-model-per-dataset paradigm to their unified approach.

**Requested Changes:**

I believe the following will strengthen the work:

1. It will be good to show some initial experiments on multi-variate settings as well as it will help validate the efficacy on whether this method works in those settings.

2. Adding some experiments which show how training + inference time and memory scale with number of memory items (M), number of retrieved items (K), and sequence length (N) will make it easier to assess the practical cost of applying the method to larger memories or longer input windows.

3. It will be good to show a clean ablation on patch-level vs point-level memory with the same pre-trained backone and the same multi-domain training setting. It will help strengthen the arguement of using patch-level memory.

---

> ### Author Response · Authors · 2026-06-16
>
> We appreciate Reviewer MbtF’s insightful feedback on the scope and practical aspects of the proposed method. The suggestions encouraged us to more clearly define the univariate setting, add runtime scaling analysis, and provide additional evidence for the benefit of patch-level memory representations.
>
> >1. The paper only considers univariate datasets for evals, it will be good to show some initial results on multivariate setting as well.
>
> We agree that extending the evaluation to multivariate time series is an important direction. In this work, however, we focus on univariate time series anomaly detection, and the proposed patch-based memory module has been designed and evaluated under this setting. A direct extension to multivariate data would require additional design choices, such as how to model cross-channel dependencies and how to organize memory items across temporal patches and variables, which are beyond the scope of the current study.
>
> To avoid overstating the applicability of the method, we have clarified the scope of our work as univariate time series anomaly detection. In the revised manuscript, **the contribution summary at the end of the Introduction** now explicitly states that MOMEMTO is developed for univariate time series anomaly detection and evaluated on 23 univariate benchmark datasets. In addition, **Section 5 (Conclusion)** has been revised to acknowledge the univariate scope as a limitation and to discuss multivariate extension as an important direction for future work.
>
>
> >2. The paper lacks scaling experiments showing how training / inference time, and memory usage change as the memory size and number of retrieved memory items increase. Table 4 provides useful point estimates for one configuration, but it does not show how the method scales with important hyperparams like memory size and number of retrieved items. This makes it harder to assess the practical cost of applying the method to larger memories or longer input windows.
>
> To address this point, we have added a runtime scaling analysis in **Appendix C.9 (Runtime scaling with memory and sequence length)**
>
> In this section, we measure per-batch training time, inference time, and peak GPU memory while varying the number of memory items (M), the number of retrieved memory items (K), and the number of input patches (N). All measurements have been conducted under the same full fine-tuning setting with a fixed batch size and three repeated runs.
>
> The added results show that increasing (M) from 4 to 128 and (K) from 1 to 16 has only a limited effect on per-batch runtime within the tested range. In contrast, the clearest scaling trend appears with respect to the number of input patches (N), where increasing (N) from 8 to 64 increases training time and peak GPU memory. This additional analysis clarifies the practical computational cost of the proposed memory module under larger memories and longer input windows.
>
> >3. The paper lacks clean ablations on comparing point-level and patch level with the same pre-trained backbone. In Table 14 they compare MEMTO and MOMEMTO but the latter has a different pretrained backbone so it is hard to verify whether patch-level memory is more beneficial than point-level
>
> Since existing point-level memory methods are structurally incompatible with a pre-trained patch-based backbone, a direct comparison under the same pre-trained backbone is not straightforward.
>
> As an alternative, we have added a patch-length sensitivity experiment using the same TSFM backbone architecture initialized from scratch in **Appendix C.7 (Patch Size Sensitivity)**. In this experiment, we reduce the patch size down to (L=1), corresponding to a point-level setting. Through this controlled comparison, we aim to indirectly examine the benefit of the patch-based structure.
>
> | Patch length $L$ |    AUC-PR |   AUC-ROC |
> | ---------------: | --------: | --------: |
> |                1 |     28.87 |     67.52 |
> |                2 |     30.71 |     70.24 |
> |                4 |     32.69 |     70.68 |
> |                8 |     33.63 |     72.17 |
> |           **16** | **34.08** | **72.64** |
>
> As shown in the above table, the point-level setting ((L=1)) yields the lowest performance, while increasing the patch length improves both AUC-PR and AUC-ROC. Although this is an indirect comparison, the result suggests that patch-level representations better capture local temporal structure and are more effective for memory-guided reconstruction in our setting.

---

### Review · Reviewer_nmyJ · 2026-06-02

**Summary Of Contributions:**

This paper proposes MOMEMTO, a patch-based memory-augmented time series foundation model for unsupervised time series anomaly detection. The method builds upon the MOMENT backbone and introduces a patch-based memory module to store representative normal patterns, aiming to mitigate the over-generalization problem of reconstruction-based anomaly detection models. The paper also adopts a multi-domain training strategy, where a single model is jointly fine-tuned across multiple univariate time series datasets. Experiments on the TSB-AD-U benchmark show that the proposed method improves AUC and VUS metrics compared with several classical, deep learning, and foundation-model baselines, and the paper further provides ablation studies, few-shot analysis, memory utilization analysis, and efficiency comparisons.

**Additional Comments:**

My overall recommendation is to reject the submission in its current form.

**Audience:**

Yes

**Audience Explanation:**

The topic of applying time series foundation models to anomaly detection is timely and relevant, and some readers in the TMLR community may be interested in the empirical findings on memory-augmented reconstruction and multi-domain fine-tuning. The reported results on the TSB-AD-U benchmark may also be useful for researchers working on time series anomaly detection and foundation models. However, this interest does not necessarily imply that the paper is ready for acceptance, since the current submission still has important weaknesses in novelty, evidence, ablation analysis, and experimental rigor.

**Broader Impact Concerns:**

I do not identify major broader impact concerns for this work. The paper focuses on time series anomaly detection and does not appear to introduce direct ethical or societal risks beyond those commonly associated with automated anomaly detection systems. However, if the method is intended for safety-critical cyber-physical systems, the authors may need to discuss the potential impact of false positives and false negatives, especially given that the current evaluation is mainly conducted on univariate benchmark datasets.

**Claims And Evidence:**

No

**Claims Explanation:**

While the paper reports improved anomaly detection performance, the evidence does not fully support its main claims. In particular, the claim that the patch-based memory module mitigates over-generalization is not directly validated, since the paper mainly relies on downstream AUC/VUS metrics rather than analyzing how the reconstruction behavior changes for anomalous samples. The ablation studies are also incomplete and do not sufficiently isolate the contributions of key architectural choices such as the memory-refined query, the Top-K memory selection/update mechanism, and the overall memory gate design. In addition, some experimental settings, such as window sizes and preprocessing strategies, appear to differ across baselines, which makes the comparisons less convincing. Therefore, I do not believe the claims are supported by sufficiently accurate, clear, and convincing evidence.

**Requested Changes:**

1. The current method appears to mainly combine an existing time series foundation model with a memory-guided reconstruction module, and the paper should more clearly explain what is technically new compared with MOMENT, MEMTO, and other memory-based anomaly detection methods.
2. The current evidence mainly relies on final AUC/VUS metrics, but it does not clearly show how the reconstruction behavior changes for anomalous samples after introducing the memory module.
3. The current discussion based on dense associative memory is mostly an intuitive analogy and does not provide formal guarantees that the proposed module can reduce over-generalization or improve anomaly separability. The authors should either provide a more rigorous analysis or present this part more modestly as intuition rather than theory.
4.  The paper should explicitly define the query, key, and value roles in the attention operations, clarify the dimensions of memory items and patch-level queries, and explain how the Top-K memory selection is performed. Some current equations and algorithmic descriptions are difficult to follow.
5. In particular, the authors should compare decoding from only the raw query, only the memory-refined query, and their concatenation. Additional comparisons with simpler alternatives such as random memory, frozen memory, or a generic bottleneck would also help verify whether the improvement truly comes from the proposed memory mechanism.
6. The authors should clarify the experimental alignment with baselines. It should be explicitly stated whether all methods use comparable window sizes, preprocessing pipelines, and fine-tuning protocols. Results averaged over multiple random seeds with standard deviations should also be reported to support statistical reliability. In addition, Figure 1 should be redesigned. The current diagram is not sufficiently clear and contains visual inconsistencies, such as using the summation symbol $\oplus$ while the text describes feature concatenation. The figure should more clearly illustrate the full data flow, including memory selection, memory update, query update, reconstruction, and final anomaly scoring.

---

> ### Author Response · Authors · 2026-06-16
>
> We sincerely thank Reviewer nmyj for the careful reading and constructive suggestions. The comments helped us clarify the relationship between MOMEMTO and existing methods, improve the presentation of the patch-based memory module, and strengthen the analysis of reconstruction behavior.
>
>
> >1. The current method appears to mainly combine an existing time series foundation model with a memory-guided reconstruction module, and the paper should more clearly explain what is technically new compared with MOMENT, MEMTO, and other memory-based anomaly detection methods
>
> To address this concern, we have revised the **final paragraph of Section 2 (Related Work)** to better position the proposed method relative to existing memory-based anomaly detection methods, including MEMTO and H-PAD. The revised discussion clarifies the technical differences introduced by adapting memory-guided reconstruction to the TSFM setting, such as using pre-trained patch-level representations, selective data-driven memory updates, multi-domain training, and reconstruction-error-based anomaly scoring.
>
> In addition, we have revised **Section 3.1.2 (Patch-based Memory Module)** to explicitly state that, unlike MEMTO’s point-level memory, the proposed module is designed to align with the patch-based encoder representations of MOMENT. Together, these revisions clarify that the TSFM backbone and memory mechanism are complementary components, rather than a simple combination of existing methods.
>
> >2. The current evidence mainly relies on final AUC/VUS metrics, but it does not clearly show how the reconstruction behavior changes for anomalous samples after introducing the memory module.
>
> To address this point, we have added a reconstruction-behavior analysis in the **"Over-generalization alleviation” paragraph of Section 4.2 (Model Analysis)**. The added controlled synthetic analysis compares MOMENT, our TSFM backbone, with MOMEMTO on anomalous samples and illustrates how the patch-based memory module can alter the reconstruction tendency beyond aggregate AUC/VUS metrics.
>
> We have also added **Appendix C.1 (Qualitative Case Analysis of MOMENT and MOMEMTO)**, which compares the input series, reconstructed outputs, and anomaly-score profiles of both models in representative success and failure cases. These qualitative results further support that similar reconstruction patterns appear in benchmark time series.
>
> >3. The current discussion based on dense associative memory is mostly an intuitive analogy and does not provide formal guarantees that the proposed module can reduce over-generalization or improve anomaly separability. The authors should either provide a more rigorous analysis or present this part more modestly as intuition rather than theory.
>
> We agree with the reviewer’s comment. The dense associative memory discussion was intended only as an intuition for the observed concentration of memory usage on a subset of memory items, not as a formal guarantee of reduced over-generalization or improved anomaly separability.
>
> To avoid overstating the theoretical claim, we have removed the dense associative memory discussion. Instead, we have strengthened the empirical analysis in **Appendix C.4 (Number of Memory Items)**, which reports memory-size sensitivity results and visualizes memory-access patterns across different memory capacities. We have also added a cross-reference in the **data-driven memory utilization analysis in Section 4.2 (Model Analysis)** to direct readers to this additional appendix analysis.
>
> > 4. The paper should explicitly define the query, key, and value roles in the attention operations, clarify the dimensions of memory items and patch-level queries, and explain how the Top-K memory selection is performed. Some current equations and algorithmic descriptions are difficult to follow
>
> To address this point, we have revised **Section 3.1.2 (Patch-based Memory Module)** to make the memory mechanism easier to follow. Specifically, we added Table 1 (Notation summary for MOMEMTO), which summarizes the dimensions and descriptions of the main variables used in the memory module. We also reorganized the explanation into separate paragraphs for Memory initialization, Memory alignment, Memory selection, Memory update, Query update, Query alignment, and Memory dynamics, rather than presenting the memory process in a condensed form.
>
> In addition, we revised Algorithm 2 in **Appendix B.4 (Algorithms)** to match this step-by-step description. For example, the revised Memory selection paragraph and Algorithm 2 now explicitly describe the sequence-level similarity computation, Top-K retrieval from raw scores, and softmax normalization over only the selected memory items.

---

> ### Author Response · Authors · 2026-06-16
>
> >5. In particular, the authors should compare decoding from only the raw query, only the memory-refined query, and their concatenation. Additional comparisons with simpler alternatives such as random memory, frozen memory, or a generic bottleneck would also help verify whether the improvement truly comes from the proposed memory mechanism
>
> Following the reviewer’s suggestion, we have added **Appendix C.5 (Decoder Input Ablation)** and Table 15 to examine the effect of decoder input representations. Specifically, Table 15 compares decoding from the original query $\mathbf{q}$, the memory-refined query $\tilde{\mathbf{q}}$, and their concatenation $[\mathbf{q};\tilde{\mathbf{q}}]$, showing how each representation affects detection performance. We have also added a sentence in the **Ablation study paragraph of Section 4.2 (Model Analysis)** to refer readers to this additional decoder-input ablation. The frozen-memory setting is reported as the “No update” baseline in **Table 4 of Section 4.2**, under the Data-driven memory utilization analysis.
>
> (Table 15 in Appendix C.5)
> | Decoder input | AUC-PR | AUC-ROC |
> |---|---:|---:|
> | Original query only ($\mathbf{q}$) | 29.58 | 69.33 |
> | Memory-refined query only ($\tilde{\mathbf{q}}$) | 32.58 | 69.54 |
> | Concatenation ($[\mathbf{q}; \tilde{\mathbf{q}}]$) | 36.35 | 74.83 |
> >6. The authors should clarify the experimental alignment with baselines. It should be explicitly stated whether all methods use comparable window sizes, preprocessing pipelines, and fine-tuning protocols. Results averaged over multiple random seeds with standard deviations should also be reported to support statistical reliability. In addition, Figure 1 should be redesigned. The current diagram is not sufficiently clear and contains visual inconsistencies, such as using the summation symbol  while the text describes feature concatenation. The figure should more clearly illustrate the full data flow, including memory selection, memory update, query update, reconstruction, and final anomaly scoring
>
> To address this point, we have clarified the experimental alignment with baseline methods in the **Experimental setup and Implementation details paragraphs of Section 4 (Experiments)**, as well as in **Appendix B.3 (Baselines and Hyperparameter Settings)**. The revised descriptions make the preprocessing pipeline, train-only normalization, subsequence/window generation, and fine-tuning protocols more explicit. We have also updated **Table 2 in Section 4.1 (Main Results)** to report the mean and standard deviation over five random seeds. In addition, to support a fairer comparison among TSFM baselines, we have added **Appendix C.6 (Additional Comparison with Similar Input Window Lengths) and Table 16**, which report results under comparable input window lengths.
>
> (Table 16 in Appendix C.6)
> | Model | Win. len. | AUC-PR | AUC-ROC |
> |---|---:|---:|---:|
> | Chronos$_{md}$ | 100 | 26.43 | 65.48 |
> | TimesFM$_{md}$ | 96 | 26.67 | 66.95 |
> | MOMENT$_{md}$ | 96 | 26.71 | 67.27 |
> | DADA$_{md}$ | 100 | 33.08 | 70.04 |
> | **MOMEMTO$_{md}$** | **96** | **34.90** | **73.00** |
>
>
> We have redesigned **Figure 1 (Architecture of MOMEMTO)** to more clearly illustrate the overall data flow of the proposed method. The revised figure presents the procedure in a more step-by-step manner, including memory selection, memory update, query update, reconstruction, and final anomaly scoring, and removes the previous visual inconsistency between feature concatenation and the summation symbol.

---

### Author Response · Authors · 2026-06-16

We thank the reviewers for their valuable and constructive feedback. We have re-uploaded the revised manuscript, with all corresponding changes highlighted in blue. We incorporated the revisions described in our responses and summarized the main changes below.

## Major Changes

### Section 1: Introduction and Scope
- Clarified that the paper focuses on univariate time series anomaly detection and explicitly states that MOMEMTO is evaluated on 23 univariate benchmark datasets.
- Softened broad claims about general applicability so that they are tied to the demonstrated benchmark, metrics, and experimental setting.

### Section 2: Related Work
- Revised the discussion of memory-based anomaly detection methods, including MEMTO and H-PAD, to better position MOMEMTO relative to existing approaches.
- Clarified how MOMEMTO adapts memory-guided reconstruction to the TSFM setting through pre-trained patch-level representations, selective data-driven memory updates, multi-domain training, and reconstruction-error-based scoring.

### Section 3: Methodology
- Redesigned Figure 1 to show the full MOMEMTO data flow, including memory selection, memory update, query update, reconstruction, and anomaly scoring.
- Substantially revised Section 3.1.2 to make the patch-based memory module easier to follow.
- Added a notation table defining the main variables, tensor shapes, and descriptions of queries, memory items, retrieval weights, update representations, and memory-refined queries.
- Reorganized the memory mechanism into step-by-step descriptions: initialization, alignment, selection, update, query update, query alignment, and memory dynamics.
- Clarified Top-K memory selection, including sequence-level similarity computation, retrieval from raw scores, and softmax normalization over the selected items.
- Added memory-state semantics explaining that memory items are persistent state variables, not parameters directly optimized by gradients. We also clarified update persistence across batches, gradient flow, and fixed-memory inference.
- Updated Algorithm 2 in Appendix B.4 accordingly.

### Section 4: Experiments and Results
- Clarified the evaluation protocol, including train/test splits, train-only normalization, window generation, fine-tuning settings, and metric aggregation.
- Clarified that metrics are computed independently for each of the 350 Eval time series and then averaged.
- Replaced the previous one-model-per-series comparison with a one-model-per-domain setting. The 350 Eval time series are grouped into 32 domains, and relevant baselines were rerun under this revised setting.
- Updated Table 2 and Figure 2 to report means and standard deviations over five random seeds.

### Section 4.2: Model Analysis
- Added reconstruction-behavior analysis comparing MOMENT and MOMEMTO on anomalous inputs to examine over-generalization more directly.
- Added a cross-reference from the data-driven memory utilization analysis to Appendix C.4 for additional memory-size and memory-access analysis.

### Appendix: Additional Analyses
- Added Appendix C.1, a qualitative case analysis comparing input series, reconstructions, and anomaly-score profiles for representative success and failure cases.
- Strengthened Appendix C.4 by adding memory-size sensitivity analysis and memory-access visualizations. We also removed the dense associative memory discussion to avoid presenting an intuitive analogy as a formal guarantee.
- Added Appendix C.5, a decoder input ablation comparing the original query, memory-refined query, and their concatenation.
- Added Appendix C.6, an additional TSFM comparison under similar input window lengths.
- Added Appendix C.7, patch-size sensitivity analysis including a point-level proxy with patch length L = 1.
- Revised Appendix C.8 to report training/inference time under the 32-domain setting and added Appendix C.9 for runtime scaling with respect to memory size, retrieved memory items, and input patches.

### Section 5: Conclusion
- Revised the conclusion to acknowledge the univariate scope as a limitation and discuss multivariate extension as future work.

## Minor Changes
- Improved terminology consistency for "dataset," "domain," "sub-domain," and "time series."
- Softened broad performance statements and tied them to the specific benchmark, metrics, and settings.
- Updated captions, cross-references, and explanatory text to improve readability and reproducibility.

We sincerely appreciate the reviewers' constructive feedback and would be happy to address any further comments or questions.